# PRACTICAL ESTIMATION OF THE OPTIMAL CLASSIFICATION ERROR WITH SOFT LABELS AND CALIBRATION

**Ryota Ushio**[1,2]**, Takashi Ishida**[2,1]**, Masashi Sugiyama**[2,1]
[1] The University of Tokyo, Tokyo, Japan
[2] RIKEN AIP, Tokyo, Japan
{ushio@ms.,ishida@,sugiyama@}k.u-tokyo.ac.jp

## ABSTRACT

While the performance of machine learning systems has experienced significant improvement in recent years, relatively little attention has been paid to the fundamental question: to what extent can we improve our models? This paper provides a means of answering this question in the setting of binary classification, which is practical and theoretically supported. We extend a previous work that utilizes soft labels for estimating the Bayes error, the optimal error rate, in two important ways. First, we theoretically investigate the properties of the bias of the hard-label-based estimator discussed in the original work. We reveal that the decay rate of the bias is adaptive to how well the two class-conditional distributions are separated, and it can decay significantly faster than the previous result suggested as the number of hard labels per instance grows. Second, we tackle a more challenging problem setting: estimation with *corrupted* soft labels. One might be tempted to use calibrated soft labels instead of clean ones. However, we reveal that *calibration guarantee is not enough*, that is, even perfectly calibrated soft labels can result in a substantially inaccurate estimate. Then, we show that isotonic calibration can provide a statistically consistent estimator under an assumption weaker than that of the previous work. Our method is *instance-free*, i.e., we do not assume access to any input instances. This feature allows it to be adopted in practical scenarios where the instances are not available due to privacy issues. Experiments with synthetic and real-world datasets show the validity of our methods and theory. The code is available at https://github.com/RyotaUshio/bayes-error-estimation.

## 1 INTRODUCTION

It is a common practice in the field of machine learning research to assess the performance of a newly proposed algorithm using one or more metrics and compare them to the previous state-of-the-art (SOTA) performance to show its effectiveness (Neu, 2024; Int, 2025a;b). In classification, arguably the most common one is the error rate, i.e., the expected frequency of misclassification for future data.

While the SOTA performance continues to improve for a wide range of benchmarks over time, there is a limit on the prediction performance that any machine learning model can achieve, which is determined by the underlying data distribution. It is important to know this limit, or the best achievable performance. For example, if the current SOTA performance is close enough to the limit, there is no point in seeking further improvement. It is not only wasteful in terms of time and financial resources but also harmful to the environment, since large-scale machine learning models are notorious for their high energy consumption (Strubell et al., 2020; Luccioni et al., 2023). Knowing the best possible performance also provides a practical check for test-set overfitting (Recht et al., 2018; Ishida et al., 2023): if a model's score on the test set approaches or even exceeds the upper bound, it may be a signal of the model directly training on the test set.

In classification, the best achievable error rate for a given data distribution is called the *Bayes error*, and the estimation of the Bayes error has a rich history of research (Fukunaga and Hostetler, 1975; Devijver, 1985; Berisha et al., 2014; Moon et al., 2018; Noshad et al., 2019; Theisen et al., 2021;

Ishida et al., 2023; Jeong et al., 2023). In the case of binary classification, the existing approaches can be roughly categorized into two groups: the majority of estimation from instance-label pairs $(x_1, y_1), \ldots, (x_n, y_n) \in \mathcal{X} \times \{0, 1\}$ (Fukunaga and Hostetler, 1975; Devijver, 1985; Berisha et al., 2014; Moon et al., 2018; Noshad et al., 2019; Theisen et al., 2021), where $\mathcal{X}$ is the space of instances, and the recently proposed methods of estimation from *soft labels* $\eta_1, \ldots, \eta_n \in [0, 1]$ (Ishida et al., 2023; Jeong et al., 2023). A soft label is a special type of supervision that represents the posterior class probability $\eta_i := \mathbb{P}(y = 1 \mid x = x_i)$, $i \in \{1, \ldots, n\}$, and it quantifies the uncertainty of class labels associated with each instance $x_i$. The strength of the methods proposed in (Ishida et al., 2023; Jeong et al., 2023) based on soft labels is that they are *instance-free*, i.e., they do not require access to the instances $\{x_i\}_{i=1}^n$. Since the instances themselves are not used for estimation, these methods do not suffer from the curse of dimensionality even when dealing with very high-dimensional data. Moreover, the instance-free property is practically valuable since it makes the methods easy to apply to real-world problems, such as medical diagnoses, where the instances are often inaccessible due to privacy concerns. However, these methods have a crucial limitation: they assume that we have direct access to *clean* soft labels $\eta_i = \mathbb{P}(y = 1 \mid x = x_i)$, which only an oracle would know. Ishida et al. (2023) also discussed a scenario where each soft label $\eta_i$ is approximated by an average $\widehat{\eta}_i = \frac{1}{m} \sum_{j=1}^m y_i^{(j)}$ of $m$ hard labels $y_i^{(1)}, \ldots, y_i^{(m)}$ per instance. They showed that, for a fixed number $n$ of samples, the bias of the resulting estimator approaches zero as $m$ tends to infinity. However, their bound on the bias is prohibitively large for practical values of $m$, and thus the theoretical guarantee for this estimator is weak.

Another issue is *labeling distribution shift*. For example, whereas the images in the original CIFAR-10 dataset (Krizhevsky, 2009) can be regarded as though they were annotated before they were downscaled, the images in the CIFAR-10H dataset (Peterson et al., 2019) were annotated after downscaling, making the task more challenging and thus increasing label uncertainty. [1] Given that the Bayes error can be interpreted as the average label uncertainty, we will get an unreasonably high estimate of the Bayes error if we just plug the soft labels in

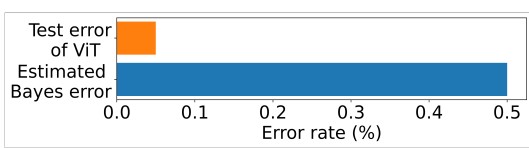

Figure 1: The Bayes error estimated with the method of Ishida et al. (2023) is larger than the test error of a Vision Transformer (Dosovitskiy et al., 2021).

CIFAR-10H into their estimator, as shown in Fig. 1. In general, a similar issue can arise due to subjectivity of human soft labelers, or the bias of using large language models (LLMs) as annotators (Gilardi et al., 2023; Tjuatja et al., 2024). Recent work has explored a range of techniques to obtain soft labels and confidence scores from LLMs, but constructing a high-quality soft label remains to be a challenge (Xie et al., 2024; Kadavath et al., 2022; Argyle et al., 2023). This distortion issue due to the distribution shift was also mentioned by Ishida et al. (2023), but no solution was shown in their paper.

**Contribution of this paper** We extend the previous work by Ishida et al. (2023) that utilizes soft labels for estimating the Bayes error, the optimal error rate, in two important ways.

First, we deepen the theoretical understanding of the bias of the hard-label-based estimator discussed in the original work. Specifically, we show that the decay speed of the bias depends on how well the two class-conditional distributions are separated, and it can approach zero significantly faster than the previous result suggested as the number $m$ of hard labels per instance grows.

Second, we discuss a new, more challenging problem of estimation from *corrupted* soft labels. In this scenario, we are given a distorted version of the clean soft labels. The distortion can arise from a shifted labeling distribution or subjectivity of human/LLM soft labelers. It reflects many real-world problems including the estimation of best achievable performances for benchmark datasets and esti-

---

[1]For more information, CIFAR-10 was curated as follows. First, images for each class were collected by searching for the class label or its hyponym on the Internet. Second, the images were downscaled to $32 \times 32$. Finally, human labelers were asked to filter out mislabeled images. On the other hand, CIFAR-10H is a soft-labeled version of CIFAR-10, i.e., it consists of 10,000 test images of the CIFAR-10 test set along with their soft labels. Each soft label was obtained as an average of 47–63 hard labels, which were collected by asking human labelers to answer which class the downscaled image belongs to. As a result, the labeling processes are significantly different between these two datasets.

mation from soft labels obtained from subjective confidence. One might be tempted to use calibrated soft labels in place of clean ones. However, we reveal that *calibration guarantee is not enough*, i.e., even perfectly calibrated soft labels can result in a substantially inaccurate Bayes error estimate, which highlights the importance of choosing appropriate calibration algorithms. Then, we show that a classical calibration algorithm called *isotonic calibration* (Zadrozny and Elkan, 2002) can provide a statistically consistent estimator as long as the original soft labels are correctly ordered.

## 2 FINE-GRAINED ANALYSIS OF THE BIAS

We first explain the preliminaries for this section and then present our main results.

### 2.1 PRELIMINARIES

**Formulation and notations**    Let $\mathcal{X} \subset \mathbb{R}^d$ and $\mathcal{Y}$ be the spaces of instances and output labels, respectively. In this paper, we confine ourselves to binary classification problems and thus we set $\mathcal{Y} = \{0, 1\}$. Classes 1 and 0 are also called the positive and negative classes, respectively. Let $\mathbb{P}$ be a distribution over $\mathcal{X} \times \mathcal{Y}$ and $(x, y)$ be a pair of random variables following $\mathbb{P}$, where $x \in \mathbb{R}^d$ is an input instance and $y \in \{0, 1\}$ is the corresponding class label. Throughout this paper, $x_1, \ldots, x_n$ are $n$ i.i.d. samples drawn from the marginal distribution $\mathbb{P}_{\mathcal{X}}$ over the input space $\mathcal{X}$. We denote by $\mathbb{P}_1$, $\mathbb{P}_0$ two class-conditional distributions over the input space $\mathcal{X}$ given $y = 1$ and $y = 0$, respectively. Note that $\mathbb{P}_{\mathcal{X}} = \theta \mathbb{P}_1 + (1 - \theta)\mathbb{P}_0$, where $\theta := \mathbb{P}(y = 1) \in (0, 1)$ is the base rate. $\eta(x)$ is the *clean soft label* for an instance $x$, i.e., the posterior probability $\mathbb{P}(y = 1 \mid x)$ of $y = 1$ given $x$. The expectation and the variance with respect to the marginal $\mathbb{P}_{\mathcal{X}}$ are denoted by $\mathbb{E}$ and $\mathrm{Var}$, respectively.

For any positive integer $n$, we denote $[n] := \{1, \ldots, n\}$. An indicator function is denoted by $\mathbb{1}[\cdot]$. Given a sequence of $n$ random variables $z_1, \ldots, z_n$, we use a shorthand $z_{1:n} = (z_1, \ldots, z_n)$. For $\mu \in \mathbb{R}^d$ and $\Sigma \in \mathbb{R}^{d \times d}$, $N(\mu, \Sigma)$ is the Gaussian distribution with mean $\mu$ and covariance $\Sigma$.

**Estimating the best possible performance with soft labels**    Among the most commonly used performance measures would be the error rate. The error rate of a classifier $h : \mathcal{X} \to \mathcal{Y}$ is defined as $\mathrm{Err}(h) := \mathbb{E}_{(x,y)\sim\mathbb{P}}[\mathbb{1}[y \neq h(x)]]$, where $(x, y)$ is a test instance-label pair drawn independently of training data. The best possible error rate $\mathrm{Err}^* := \inf_{h:\mathcal{X}\to\mathcal{Y}} \mathrm{Err}(h)$ is called the *Bayes error* (Mohri et al., 2018).[2]

Recall that $\eta(x) := \mathbb{P}(y = 1 \mid x)$. Ishida et al. (2023) proposed a direct approach to estimating the Bayes error $\mathrm{Err}^*$ assuming access to the soft labels $\{\eta(x_i)\}_{i=1}^n$ rather than instance-label pairs $\{(x_i, y_i)\}_{i=1}^n$. Its derivation is outlined as follows. First, it is well-known that the Bayes error can be expressed as $\mathrm{Err}^* = \mathbb{E}_{x\sim\mathbb{P}_{\mathcal{X}}}[\min\{\eta(x), 1 - \eta(x)\}]$ (Cover, 1968). Replacing the expectation with a sample average over $\{x_i\}_{i=1}^n$, they obtained an unbiased estimator

$$\widehat{\mathrm{Err}^*}(\eta_{1:n}) := \frac{1}{n}\sum_{i=1}^n \min\{\eta_i, 1 - \eta_i\}, \tag{1}$$

where $\eta_i := \eta(x_i)$. It is also statistically consistent, or more specifically, for any $\delta \in (0, 1)$, with probability at least $1 - \delta$, we have $\left|\widehat{\mathrm{Err}^*}(\eta_{1:n}) - \mathrm{Err}^*\right| \leq \sqrt{\frac{\log(2/\delta)}{8n}}$.

In practical terms, the *instance-free* nature of this method exhibits considerable advantages over existing methods described in Appendix A despite its simplicity. It can be applied to settings where input instances themselves are unavailable, e.g., due to privacy issues. On the other hand, one of the crucial drawbacks of this method is that clean soft labels are usually inaccessible in practice. Therefore, we have no choice but to substitute some estimates for them. Ishida et al. (2023) also considered a setting where $\eta_i$ is approximated by an average of hard labels $\widehat{\eta}_i := \frac{1}{m}\sum_{j=1}^m y_i^{(j)}$, where $y_i^{(1)}, \ldots, y_i^{(m)}$ are $m$ hard labels each of which is drawn independently from the posterior distribution of the class labels given the instance $x_i$. In practice, $y_i^{(1)}, \ldots, y_i^{(m)}$ could be collected by asking $m$ different human labelers to answer whether $x_i$ belongs to class 1. CIFAR-10H (Peterson et al.,

---

[2]The infimum is taken over all measurable functions.

2019) and Fashion-MNIST-H (Ishida et al., 2023) are examples of a dataset constructed as such. By plugging $\widehat{\eta}_i$ in place of $\eta_i$, they obtained the estimator $\widehat{\mathrm{Err}}^*(\widehat{\eta}_{1:n}) = \frac{1}{n} \sum_{i=1}^{n} \min\{\widehat{\eta}_i, 1 - \widehat{\eta}_i\}$. They showed that the bias is bounded as

$$\left| \mathbb{E}\left[ \widehat{\mathrm{Err}}^*(\widehat{\eta}_{1:n}) \right] - \mathrm{Err}^* \right| \leq \frac{1}{2\sqrt{m}} + \sqrt{\frac{\log(2n\sqrt{m})}{m}}, \qquad (2)$$

and thus vanishes as $m \to \infty$ given $n$ fixed. However, the rate $\tilde{\mathcal{O}}\left(1/\sqrt{m}\right)$ is quite slow given that $m$ is typically much smaller than $n$. For example, in the case of the CIFAR-10H dataset, each image is given only around 50 hard labels. Later in Section 2.2, we will show that this rate can be significantly improved for well-conditioned distributions. In addition, the second term on the right-hand side of (2) increases as $n$ grows, which appears to be unnatural. We also show that the bias can be upper-bounded by a quantity irrelevant to $n$.

## 2.2 Main results

**Improved bound on the bias**  The following theorem provides a new bound on the bias of the estimator $\widehat{\mathrm{Err}}^*(\widehat{\eta}_{1:n})$. The proofs for all results in this section can be found in Appendix B.

**Theorem 1.** *We have*

$$- \mathop{\mathbb{E}}_{x \sim \mathbb{P}_{\mathcal{X}}}\left[ \min\left\{ \frac{L_{\mathrm{Err}}(\eta(x))}{m}, \sqrt{\frac{\pi}{2m}} \right\} \right] \leq \mathbb{E}\left[ \widehat{\mathrm{Err}}^*(\widehat{\eta}_{1:n}) \right] - \mathrm{Err}^* \leq 0, \qquad (3)$$

*where $L_{\mathrm{Err}}(q)$ is $\frac{q(1-q)}{|2q-1|}$ if $q \neq 0.5$ and $\infty$ if $q = 0.5$.*

First of all, we note that our upper bound does not contain $n$ unlike the existing result (2) by Ishida et al. (2023). More importantly, our bound (3) indeed improves upon theirs (Proposition 1). We only need a weaker version of (3) to show the improvement: $-\sqrt{\frac{\pi}{2m}} \leq \mathbb{E}\left[ \widehat{\mathrm{Err}}^*(\widehat{\eta}_{1:n}) \right] - \mathrm{Err}^* \leq 0$.

Although the $\frac{L_{\mathrm{Err}}(\eta(x))}{m}$ term in the left-hand side of (3) is unnecessary to improve the previous result (2), it provides further insights. Fig. 5 in Appendix B shows the graph of the function $L_{\mathrm{Err}}(p)$. It takes a near-zero value when $p$ is close to 0 or 1 and diverges to infinity as $p \to 0.5$. This means that $L_{\mathrm{Err}}(\eta(x))$ is large for instances $x$ close to the Bayes-optimal decision boundary $\eta(x) = 0.5$, while it is close to zero for instances far away from it. Therefore, the rate at which the bias decays can be understood as a mixture of the fast rate $1/m$ and the slower rate $1/\sqrt{m}$, whose weights are determined by how well the two classes are separated. We validate this with numerical expeimrents in Appendix B.3.

**Well-separated cases**  Here we note that real-world datasets often have well-separated classes; see, e.g., Fig. 6 in Appendix B. If the two classes are perfectly separated, the bias can decay at the fast rate $\mathcal{O}(1/m)$, as opposed to the worst cases rate $\mathcal{O}(1/\sqrt{m})$.

**Corollary 1.** *Suppose there exists a constant $c > 0$ such that $|\eta(x) - 0.5| \geq c$ holds almost surely. Then, we have*

$$-\frac{1 - 4c^2}{8cm} \leq \mathbb{E}\left[ \widehat{\mathrm{Err}}^*(\widehat{\eta}_{1:n}) \right] - \mathrm{Err}^* \leq 0. \qquad (4)$$

The assumption of Corollary 1 is satisfied by, for example, the following distribution.

**Example 1** (Perfectly separated distributions with label noise)**.**  Consider two continuous distributions $\mathcal{F}_0, \mathcal{F}_1$ over $\mathcal{X}$ with disjoint supports. An instance-label pair $(x, y)$ is generated as follows. First, an index $k$ is selected from $\{0, 1\}$ with equal probability. Given $k$, $x$ and $y$ are generated conditionally independently as follows: (i) The instance $x$ is sampled from $\mathcal{F}_k$. (ii) The label $y$ is set to $1 - k$ with conditional probability $\nu$ and $k$ with conditional probability $1 - \nu$, where $\nu \neq 0.5$. Then, the assumption of Corollary 1 is satisfied for $c = |\nu - 0.5|$.

**Computable bound for general cases**  Our results so far (Theorem 1 and Corollary 1) provide tigher bounds and a more detailed perspective on the bias. However, a downside of those results is that, in order to compute the numerical values of the lower bounds directly, we need to know some characteristics of the data distribution that might not be available in most practical scenarios. The good news is that we can still derive a computable bound that only requires an upper bound on the Bayes error, e.g., the error rate of the SOTA model, from Theorem 1.

**Corollary 2.** *Assume that* $\mathrm{Err}^* \leq E$. *Then, we have* $-B(E, m) \leq \mathbb{E}\left[\widehat{\mathrm{Err}}^*(\widehat{\eta}_{1:n})\right] - \mathrm{Err}^* \leq 0$, *where*

$$B(E, m) := \inf_{t \in (0, 1/2)} \left( \frac{t(1-t)}{1-2t} \frac{1}{m} + \min\left\{1, \frac{E}{t}\right\} \sqrt{\frac{\pi}{2m}} \right). \tag{5}$$

The function $B(E, m)$ can be computed numerically without any information about the data distribution, except for an upper bound of the Bayes error, $E$. Fig. 2 shows the magnitude of our lower bound, $B(E, m)$, for various values of $E$ (the blue line). It also shows the existing bias bound by Ishida et al. (2023) (the orange line). This comparison demonstrates that Corollary 2 is a substantial improvement over the existing result across the entire range of the parameter $E$.

Let us take the binarized[3] CIFAR-10 test set as an example. It consists of $n = 10000$ instances, each of which has a soft label obtained as the average of around $m = 50$ hard labels from the CIFAR-10H dataset. As the parameter $E$, we can use the Vision Transformer (Dosovitskiy et al., 2021)'s empirical error rate of $0.0005$ reported by Ishida et al. (2023), which is shown by the black dashed line in Fig. 2. While the existing bound suggests the bias of the hard-label-based estimator $\widehat{\mathrm{Err}}^*(\widehat{\eta}_{1:n})$ could be as large as $0.557$, our bound reveals the estimator is not that bad; indeed, it implies the bias is never larger than $0.00276$. In this case, our result is over 200 times tighter than theirs.

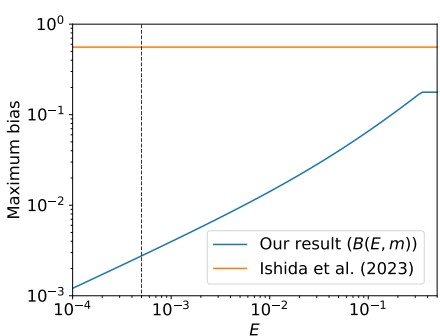

Figure 2: A comparison of our bias bound (Corollary 2) and the existing bound by Ishida et al. (2023) with $n = 10000$, $m = 50$. The dashed line indicates the test error (0.05%) of the SOTA model for the binarized CIFAR-10 dataset, which we can use as $E$ in Corollary 2. Our bound is over 200 times tighter than the existing one in this setup.

Theorem 1 also implies the estimator $\widehat{\mathrm{Err}}^*(\widehat{\eta}_{1:n})$ is statistically consistent; see Corollary 3 for details.

## 3 ESTIMATION FROM CORRUPTED SOFT LABELS

This section tackles a more challenging setting where we do *not* have access to the true posterior probability $\eta$. This setting reflects many real-world problems, such as in medical diagnosis where a doctor's subjective confidence in their decision can be regarded as a soft label, or when practitioners provide automated soft labels with LLMs in place of human annotators. However, there is no guarantee that it exactly reflects the true underlying probability.

In this section, we consider a problem setting where each instance $x_i$ is given a real number $\tilde{\eta}_i \in [0, 1]$, which is expected to approximate the clean soft label $\eta_i$ in some sense but not necessarily identical to $\eta_i$. We call $\{\tilde{\eta}_i\}_{i=1}^{n}$ *corrupted* soft labels. How can we estimate the Bayes error when only corrupted soft labels are available instead of clean ones? Estimation with a provable guarantee will be impossible without some assumption on the quality of the soft labels. Then, what guarantee can be provided under what assumption?

### 3.1 PRELIMINARIES

In addition to the preliminaries introduced in Section 2.1, we briefly review a few more required for the discussion in this section.

**Calibration**   Predicting accurate class labels is not always sufficient in classification problems. It is often crucial to obtain reliable probability estimates, especially in high-stakes applications including personalized medicine (Jiang et al., 2012) and meteorological forecasting (Murphy, 1973; DeGroot and Fienberg, 1983).

---

[3]See Section 4.1 for details.

A popular notion to capture the quality of probability estimates is *calibration*. A probabilistic classifier $c$ is said to be *well-calibrated* if the predicted probabilities closely match the actual frequencies of the class labels (Kull et al., 2017), i.e., $c(x) = \mathbb{E}[y \mid c(x)]$ almost surely. While one might hope that a perfectly calibrated output $c(x)$ matches the posterior probability $\mathbb{E}[y \mid x] = \mathbb{P}(y = 1 \mid x)$, it is not necessarily true. Indeed, calibration is a weaker notion and just a necessary condition for $c(x) = \mathbb{E}[y \mid x]$. For example, even a constant predictor $c(x) \equiv \mathbb{P}(y = 1)$ is well-calibrated [4] although it can be far from $\mathbb{P}(y = 1 \mid x)$.

It is known that many machine learning models, including modern neural networks, are not calibrated out of the box (Zadrozny and Elkan, 2001; Guo et al., 2017). Therefore, their outputs have to be recalibrated in post-processing, and various methods have been proposed to achieve this goal. They can be roughly categorized into two groups, namely parametric and nonparametric methods. The former includes *Platt scaling* (Platt, 1999), also known as *logistic calibration* (Kull et al., 2017), and *beta calibration* (Kull et al., 2017). Among the latter category are *histogram binning* (Zadrozny and Elkan, 2001) and *isotonic calibration* (Zadrozny and Elkan, 2002). Each of these methods requires a dataset $\{(c_i, y_i)\}_{i=1}^n$ to obtain a function that takes the output of an uncalibrated predictor $c$ and transforms it into a reliable probability estimate. This function is sometimes called a *calibration map* (Kull et al., 2017; 2019). Here, $c_i = c(x_i)$ is the output of $c$ for an instance $x_i$, and $y_i$ is the corresponding class label. To avoid overfitting, each $(x_i, y_i)$ needs to be sampled independently of the training set used to obtain the predictor $c$.

**Isotonic calibration** Here, we briefly describe the algorithm of isotonic calibration (Zadrozny and Elkan, 2002), arguably one of the most commonly used nonparametric recalibration methods, as it plays an important role in this paper. Suppose a dataset $\{(c_i, y_i)\}_{i=1}^n$ is given. The algorithm proceeds as follows. First, the dataset $(c_1, y_1), \ldots, (c_n, y_n)$ is reordered into $\left(c_{(1)}, y_{(1)}\right), \ldots, \left(c_{(n)}, y_{(n)}\right)$ so that the resulting sequence $c_{(1)}, \ldots, c_{(n)}$ of outputs becomes non-decreasing. Then, we find a non-decreasing sequence $0 \leq c'_{(1)} \leq \cdots \leq c'_{(n)} \leq 1$ such that it minimizes the squared error $\frac{1}{n} \sum_{i=1}^n \left(y_{(i)} - c'_{(i)}\right)^2$. Finally, for each $i \in [n]$, $c'_{(i)}$ is assigned as the calibrated version of $c_{(i)}$. This procedure is a special case of *isotonic regression*, one of the most well-studied shape-constrained regression problems.

### 3.2 Proposed method

We propose a simple approach where we first calibrate the corrupted soft labels and then plug them into the formula (1) for clean soft labels. Although calibration was originally developed for transforming the output scores of classifiers into reliable probability estimates, here we suggest using it for corrupted soft labels. We assume that, for each $i \in [n]$, we are given a corrupted soft label $\tilde{\eta}_i \in [0, 1]$ and a single hard label $y_i \in \{0, 1\}$ sampled from the true posterior distribution $\mathbb{P}(y \mid x = x_i)$. We use the hard labels $\{y_i\}_{i=1}^n$ to calibrate the soft labels $\{\tilde{\eta}_i\}_{i=1}^n$ using some calibration algorithm $\mathcal{A}$. We write $\widehat{\eta}_i^{\mathcal{A}}$ to represent the resulting calibrated soft labels. Finally, we estimate the Bayes error $\mathrm{Err}^*$ by $\widehat{\mathrm{Err}}^* \left(\widehat{\eta}_{1:n}^{\mathcal{A}}\right) = \frac{1}{n} \sum_{i=1}^n \min \left\{\widehat{\eta}_i^{\mathcal{A}}, 1 - \widehat{\eta}_i^{\mathcal{A}}\right\}$.

However, as we mentioned earlier, even perfect calibration does not necessarily imply that the resulting soft labels are accurate estimates of the clean soft labels. A simple example illustrates this limitation:

**Example 2.** Consider drawing instances from a mixture of two distributions over $\mathcal{X}$ with disjoint supports, and let us set the mixture rate $\theta$ to be $0.5$. The true Bayes error is trivially $0$. If $\mathcal{A}$ is a calibration algorithm that produces constant soft labels $\widehat{\eta}_i^{\mathcal{A}} = \theta = 0.5$ for all $i \in [n]$, it indeed achieves perfect calibration. However, the resulting estimate of the Bayes error is $\min\{\theta, 1 - \theta\} = 0.5$, which deviates significantly from the true value $0$.

Therefore, estimation with a provable guarantee will not be possible for arbitrary calibration algorithms or without any assumptions on the soft labels. What calibration algorithm can achieve reliable estimation under what assumption? In Section 3.3, we provide the first answer to this question.

---

[4]If $c(x)$ takes the constant value $\mathbb{P}(y = 1)$ for all $x$, $\mathbb{E}[y \mid c(x)]$ is equal to $\mathbb{E}[y] = \mathbb{P}(y = 1)$ since it is a conditional expectation conditioned by a constant.

### 3.3 THEORETICAL GUARANTEE FOR ISOTONIC CALIBRATION

Here, we propose choosing isotonic calibration (Zadrozny and Elkan, 2002) as the calibration algorithm $\mathcal{A}$ and indentify a condition under which we can consistently estimate the Bayes error with our method. Specifically, we estimate the Bayes error by the following procedure:

(i) Reorder $(\tilde{\eta}_1, y_1), \ldots, (\tilde{\eta}_n, y_n)$ into $\big(\tilde{\eta}_{(1)}, y_{(1)}\big), \ldots, \big(\tilde{\eta}_{(n)}, y_{(n)}\big)$ so that the resulting sequence $\tilde{\eta}_{(1)}, \ldots, \tilde{\eta}_{(n)}$ of outputs becomes non-decreasing.

(ii) Find a non-decreasing sequence $0 \le \widehat{\eta}_{(1)}^{\mathrm{iso}} \le \cdots \le \widehat{\eta}_{(n)}^{\mathrm{iso}} \le 1$ such that it minimizes the squared error $\frac{1}{n} \sum_{i=1}^{n} \left( y_{(i)} - \widehat{\eta}_{(i)}^{\mathrm{iso}} \right)^2$. This gives us isotonic-calibrated soft labels $\widehat{\eta}_{(1)}^{\mathrm{iso}}, \ldots, \widehat{\eta}_{(n)}^{\mathrm{iso}}$.

(iii) Estimate the Bayes error as $\widehat{\mathrm{Err}^*}(\widehat{\eta}_{1:n}^{\mathrm{iso}}) = \frac{1}{n} \sum_{i=1}^{n} \min\left\{ \widehat{\eta}_{(i)}^{\mathrm{iso}}, 1 - \widehat{\eta}_{(i)}^{\mathrm{iso}} \right\}$.

The next theorem is the main theoretical result of this section, which states that we can construct a consistent estimator of the Bayes error using isotonic calibration as long as the soft labels' order is preserved. The use of isotonic regression allows us to provide a solid theoretical guarantee without making parametric assumptions on how corruption occurs. See Appendix C for the proof.

**Theorem 2.** *Suppose that there exists an increasing function $f$ such that $\tilde{\eta}_i = f(\eta_i)$ almost surely. Then, for any $\delta \in (0, 1)$, with probability at least $1 - \delta$, we have*

$$\left| \widehat{\mathrm{Err}^*}(\widehat{\eta}_{1:n}^{\mathrm{iso}}) - \mathrm{Err}^* \right| \le C \left( \frac{1}{n^{1/3}} + \sqrt{\frac{\log(1/\delta)}{n}} \right), \tag{6}$$

*where $C > 0$ is a constant.*

Note that the assumption of Theorem 2 is a relaxation of the availability of clean soft labels since we can take the identity map as $f$ when $\tilde{\eta}_i = \eta_i$. In other words, the original work by Ishida et al. (2023) assumes that we have access to the exact values of the clean soft labels, whereas our proposed method only requires the knowledge of their order.

Furthermore, our result can be extended to the case where the corruption involves random noise.

**Theorem 3.** *Assume each corrupted soft label $\tilde{\eta}_i$ is generated as $\tilde{\eta}_i = f(\eta_i) + \varepsilon_i$ where $f$ is a differentiable function such that $f' \ge c$ for some constant $c > 0$ and $\varepsilon_i$ is a zero-mean random variable with variance $\le \sigma^2$. Then, for any $\delta \in (0, 1)$, with probability at least $1 - \delta$, we have*

$$\left| \widehat{\mathrm{Err}^*}(\widehat{\eta}_{1:n}^{\mathrm{iso}}) - \mathrm{Err}^* \right| \le C' \left( \sigma + \frac{1}{n^{1/3}} + \sqrt{\frac{\log(1/\delta)}{n}} \right), \tag{7}$$

*where $C' > 0$ is a constant.*

As an example, we can think of the situation we studied in Section 2 but with corrupted labeling distribution skewed by some function $f$. For each $i = 1, \ldots, n$, the resulting posterior distribution can be seen as a Bernoulli distribution with mean $f(\eta_i)$. Then, we draw $m$ hard labels $y_i^{(1)}, \ldots, y_i^{(m)}$ from that distribution. We can then approximate the unknown soft label $\eta_i$ by the average $\frac{1}{m} \sum_{j=1}^{m} y_i^{(j)}$ of the hard labels and plug it into our estimator. In this case, the randomness over the hard labels translates to additive noise with standard deviation at most $\sigma = \frac{1}{2\sqrt{m}}$.

A limitation of Theorem 3 is that it cannot be applied to $f$ whose derivative can be arbitrarily small. Roughly speaking, it is because a small fluctuation in the function value can translate to a large deviation in the inverse function value if the function is too "flat." We have not yet reached a theoretical understanding of how much violating this assumption hurts the estimation accuracy. However, in our empirical study with synthetic data (Appendix D.2), the results suggest that our method can perform well even for such corruption functions. An interesting finding is that beta calibration (Kull et al. (2017)) performs poorly even though it is a *well-specified* parametric calibrator in our experimental setting. This fact suggests that choosing appropriate calibration methods, such as isotonic calibration, is indeed crucial in our algorithm design. See Appendix D.2 for details.

# 4 EXPERIMENTS

In this section, we present experimental results where we estimate the Bayes error of synthetic and real-world datasets using our proposed method.

## 4.1 EXPERIMENTAL SETTINGS

The methods employed in this experiment were the following: (i) `clean`: the estimator with clean soft labels, i.e., $\widehat{\mathrm{Err}}^*(\eta_{1:n})$, (ii) `hard`: the estimator with approximate soft labels obtained as averaged hard labels, i.e., $\widehat{\mathrm{Err}}^*(\widehat{\eta}_{1:n})$, (iii) `corrupted`: the estimator with corrupted soft labels, i.e., $\widehat{\mathrm{Err}}^*(\tilde{\eta}_{1:n})$, and (iv) the estimator with soft labels obtained by calibrating the corrupted soft labels, i.e., $\widehat{\mathrm{Err}}^*(\widehat{\eta}_{1:n}^{\mathcal{A}})$. We used the following calibration algorithms $\mathcal{A}$: isotonic calibration (`isotonic`; Zadrozny and Elkan, 2002), uniform-mass histogram binning (Zadrozny and Elkan, 2001) with $10, 25, 50$ and $100$ bins (`hist-10`, `hist-25`, `hist-50` and `hist-100`), beta calibration (`beta`; Kull et al. (2017)), its variants with restricted parameters (`beta-am`, `beta-ab` and `beta-a`; see Appendix D.3.2 for details), and the classic Platt scaling (`platt`; Platt (1999)). Here, `isotonic` is our proposed choice of calibration algorithm in Section 3.

We used the following synthetic and real-world datasets.

**Synthetic dataset** We generated a two-dimensional synthetic dataset of size $n = 10,000$ from a Gaussian mixture $\mathbb{P}_{\mathcal{X}} = 0.6 \cdot \mathbb{P}_0 + 0.4 \cdot \mathbb{P}_1$, where $\mathbb{P}_0 = N\left((0,0)^\top, I_2\right)$, $\mathbb{P}_1 = N\left((2,2)^\top, I_2\right)$ and $I_d$ is the $d$-by-$d$ identity matrix. We used $m = 50$ hard labels per instance in the `hard` setup. For each $i \in [n]$, we generated the corrupted version $\tilde{\eta}_i$ of the soft label $\eta_i$ by $\tilde{\eta}_i = f(\eta_i; 2, 0.7)$,

where $f(p; a, b) = \left(1 + \left(\frac{1-p}{p}\right)^{1/a} \frac{1-b}{b}\right)^{-1}$, $0 < p < 1$, $a > 0$ and $0 < b < 1$. The function $f$

is the inverse function of the two-parameter beta calibration map (Kull et al., 2017) and can express various continuous increasing transformations on the interval $(0, 1)$ depending on the parameters $a$ and $b$. Fig. 3 shows the graph of the corruption function $f$. As can be seen in the figure, it pushes probability values away from zero or one, making the soft labels "unconfident." It also distorts soft labels so that $\eta_i = 0.5$ is mapped to $\tilde{\eta}_i = f(\eta_i) = b$. Note that $f$ satisfies the assumption of Theorem 2 since it is increasing. We also explored other sets of parameters and other types of corruption; see Appendix D for details.

**CIFAR-10** The second dataset is the test set of CIFAR-10 (Krizhevsky, 2009) with soft labels taken from the CIFAR-10H dataset (Peterson et al., 2019). Since they are originally multi-class datasets, we reconstructed a binary dataset by relabeling the animal-related classes (*bird*, *cat*, *deer*, *dog*, *frog* and *horse*) as positive and the rest as negative, similarly to what Ishida et al. (2023) did in their experiments. Recall that the CIFAR-10H soft labels can be considered to be corrupted because of the mismatched labeling distributions, as we mentioned in Section 1. We compared the estimated Bayes error with the test error of a Vision Transformer (ViT) (Dosovitskiy et al., 2021) on this dataset reported by Ishida et al. (2023), which is 0.05%.

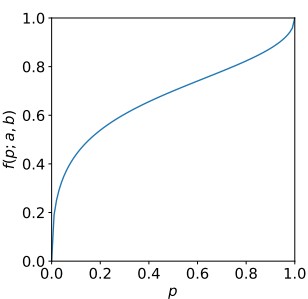

Figure 3: The corruption function $f(p; a, b)$ with parameters $a = 2$, $b = 0.7$.

**Fashion-MNIST** We also used the Fashion-MNIST dataset (Xiao et al., 2017) and its soft-labeled counterpart, Fashion-MNIST-H (Ishida et al., 2023). Following Ishida et al. (2023), we binarized the dataset by treating *T-shirt/top*, *pullover*, *dress*, *coat* and *shirt* as the positive class. The rest proceeded similarly to the CIFAR-10 experiment except that we newly trained a ResNet-18 (He et al., 2016) in place of the ViT. Details such as training parameters can be found in Appendix D.

**ChaosNLI** ChaosNLI (Nie et al., 2020; Zhou et al., 2022) is a natural language processing dataset with 100 hard labels per data point. It consists of three sub-datasets: SNLI ($n = 1,514$), MNLI ($n = 1,599$) and AbductiveNLI ($n = 1,532$).

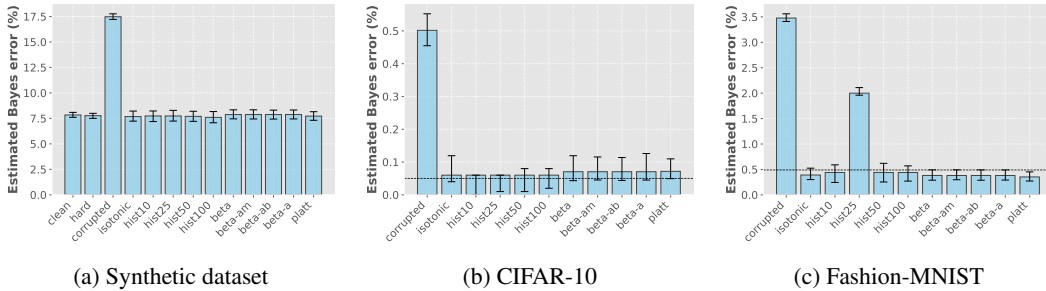

(a) Synthetic dataset                      (b) CIFAR-10                      (c) Fashion-MNIST

Figure 4: The estimated Bayes error for the synthetic dataset, CIFAR-10, and Fashion-MNIST, obtained with various methods. Error bars denote 95% bootstrap confidence intervals. For CIFAR-10, the ViT test error is indicated by a horizontal dashed line, while for Fashion-MNIST the dashed line marks the ResNet-18 test error.

**ICLR peer-review datasets** We also put together a new dataset, which consists of $n = 32,829$ instances of peer-review results for the past ICLR conferences. Peer-review can be considered as a binary classification task (accept/reject). We used our dataset to estimate the Bayes error of the ICLR reviews, which is the probability that the ideal, most competent possible reviewer mistakenly rejects a good paper or accepts a bad paper. It can be regarded as representing the inherent difficulty of the review task. For each paper submission $x_i$, we utilized the OpenReview API to retrieve the averaged score $s_i$ weighted by the confidences of the reviewers and the final decision $y_i$: accept ($y_i = 1$) or reject ($y_i = 0$). We then obtained a corrupted soft label $\tilde{\eta}_i$ for $x_i$ by normalizing the averaged score $s_i$ to fit into $[0, 1]$. We merged data from ICLR 2017–2025 to construct a dataset consisting of $n = 32,829$ examples, each of which has a corrupted soft label (i.e., normalized average score) and a single hard label (i.e., final decision) for calibration.[5]

## 4.2 ESTIMATION OF THE BAYES ERROR

Here, we estimated the Bayes error of the above-mentioned datasets using the methods described in Section 4.1. We used $1,000$ bootstrap resamples to compute a 95% confidence interval for each method. Note that, for real-world datasets, we could not experiment with the `clean` setup as clean soft labels are unavailable. Therefore, we conducted our experiment only for `corrupted`/`isotonic`/`hist-*`/`beta*`/`platt` treating the given soft labels as corrupted.[6]

**Results** Fig. 4 shows the results of the experiments with the synthetic, CIFAR-10 and Fashion-MNIST datasets. The full results including those for the ChaosNLI and ICLR peer-review datasets can be found in Appendix D.3.3. The black dashed lines in Fig. 4b and Fig. 4c indicate the test error of a classifier trained for each dataset as a reference. As expected, the underconfidence of the corrupted soft labels resulted in a severe overestimation of the Bayes error. All the calibration methods (`isotonic`, `hist-*`, `beta*` and `platt`) produced far more reasonable estimates compared with the baseline `corrupted`. For Fashion-MNIST, however, histogram binning sometimes failed to offer reasonable estimates, as can be seen in Fig. 4c. Specifically, `hist-25` resulted in a Bayes error estimate substantially larger than the ResNet's test error. This may highlight the necessity for a carefully chosen calibration method, as we mentioned in Section 3.2. On the other hand, `isotonic`, `beta*` and `platt` produced reasonable estimates in these settings.

## 4.3 COMPARING CALIBRATION ALGORITHMS USING FEEBEE

For real-world datasets, it is hard to conclude which calibration algorithm is the best just from the above results, since the true Bayes error is unknown. To address this limitation, here we compared various calibration algorithms using FeeBee (Renggli et al., 2021), a real-world evaluation frame-

---

[5]We also conducted experiments with single-year datasets (ICLR2017, . . . , ICLR2025). The results are deferred to Appendix D due to space limitations.

[6]Although we could run `hard` experiments with $m = 1$ using the hard labels from the CIFAR-10 test set, they would not produce any meaningful estimates of the Bayes error because $\min \{y, 1 - y\} = 0$ for both $y = 0$ and 1.

Table 1: FeeBee scores of calibration algorithms across real-world datasets (lower is better). The best scores for each dataset are highlighted in **bold**, and the rank within each column is shown in parentheses.

| | Dataset | | | | | |
|---|---|---|---|---|---|---|
| Algorithm | CIFAR-10 | Fashion-MNIST | SNLI | MNLI | AbductiveNLI | ICLR2017-2025 |
| isotonic | 0.00307 (3) | **0.00240 (1)** | 0.00292 (2) | 0.00147 (2) | 0.00118 (2) | 0.00013 (5) |
| hist-10 | 0.00318 (4) | 0.00250 (2) | **0.00283 (1)** | 0.00210 (5) | 0.00143 (3) | 0.00011 (3) |
| hist-25 | **0.00253 (1)** | 0.00825 (6) | 0.00356 (4) | 0.00322 (7) | 0.00362 (4) | 0.00016 (7) |
| hist-50 | 0.00322 (5) | 0.00329 (4) | 0.00520 (5) | 0.00421 (9) | 0.00558 (5) | 0.00033 (9) |
| hist-100 | 0.00329 (6) | 0.00373 (5) | 0.00714 (6) | 0.00666 (10) | 0.00813 (6) | 0.00056 (10) |
| beta | 0.02396 (8) | 0.08796 (8) | 0.01392 (8) | 0.00380 (8) | 0.05400 (9) | 0.00012 (4) |
| beta-am | 0.02401 (10) | 0.09055 (10) | 0.01452 (9) | 0.00208 (4) | 0.05204 (8) | 0.00014 (6) |
| beta-ab | 0.02335 (7) | 0.08737 (7) | 0.01476 (10) | **0.00143 (1)** | 0.04949 (7) | 0.00010 (2) |
| beta-a | 0.02400 (9) | 0.08878 (9) | 0.01370 (7) | 0.00223 (6) | 0.05537 (10) | 0.00017 (8) |
| platt | 0.00278 (2) | 0.00262 (3) | 0.00305 (3) | 0.00154 (3) | **0.00081 (1)** | **0.00001 (1)** |

work for Bayes error estimators. The key idea of FeeBee is to inject various levels of synthetic label noise and see if the estimator is able to track the resulting changes in the Bayes error; the estimator is penalized if it fails to do so. As a result, the lower the FeeBee score is, the better the estimator is. FeeBee provides a practical way to evaluate Bayes error estimators on real-world datasets without requiring knowledge of the true Bayes error rates. Appendix D.4.1 is dedicated to a more detailed review of the FeeBee framework.

For each dataset, we compared the FeeBee scores of the calibration algorithms listed in Section 4.1, namely, isotonic calibration (`isotonic`), histogram binning (`hist-10`, `hist-25`, `hist-50` and `hist-100`), beta calibration (`beta`) and its variants (`beta-am`, `beta-ab` and `beta-a`), and Platt scaling (`platt`). See Appendix D.4.2 for more details on the experimental settings.

**Results** As shown in Table 1, isotonic calibration (`isotonic`) and Platt scaling (`platt`) were among the best-performing algorithms in most datasets. Histogram binning (`hist-*`) performed well if the number of bins was appropriately chosen, but its performance was quite sensitive to this choice, implying that practitioners may need to carefully tune this hyperparameter. The performance of beta calibration and its variants (`beta*`) varied considerably depending on the dataset: they performed substantially worse than other algorithms in many datasets although they were often the best in many single-year ICLR datasets as shown in Table 2 in the appendix. We note that Platt scaling's strong empirical performance is intriguing: it implicitly assumes that the uncalibrated input scores are distributed according to a Gaussian mixture and hence are unbounded (Kull et al., 2017), yet the soft labels it is applied to are bounded in $[0, 1]$. Understanding why this mismatched assumption still yields accurate Bayes-error estimates is left as future work. Overall, these results again highlight that choosing an appropriate calibration algorithm is key to successful estimation of the Bayes error.

## 5 CONCLUSION AND DISCUSSION

We discussed the estimation of the Bayes error in binary classification. In Section 2, we significantly improved the existing bound on the bias of the hard-label-based estimator. We also revealed that the decay rate of the bias depends on how well the two class-conditional distributions are separated, and it can decay at a much faster rate than the previous result suggested. In Section 3, we tackled the challenging problem of Bayes error estimation from corrupted soft labels and proposed an estimator based on calibration. After presenting an example highlighting the importance of choosing appropriate calibration algorithms, we proved that we can construct a statistically consistent estimator using isotonic calibration as long as the original soft labels are correctly ordered. Then, our theory was validated by numerical experiments with synthetic and real-world datasets in Section 4.

Finally, we discuss possible future directions. An important one is the extension to multi-class problems. Investigating theoretical guarantees for calibration algorithms other than isotonic calibration such as Platt scaling is another interesting direction.

ACKNOWLEDGMENTS

RU was supported by the RIKEN Junior Research Associate Program, TI was supported by KAK-ENHI Grant Number 22K17946, and MS was supported by JST ASPIRE Grant Number JPM-JAP2405.

REPRODUCIBILITY STATEMENT

All the information needed to reproduce the experimental results in this paper is fully disclosed in the body text, the appendices, and the source code in the supplementary material or our GitHub repository. The body text and the appendices clearly explain the experimental settings. The supplementary material and the GitHub repository provide all the source code to run the experiments and the `README.md` file containing the instructions required to reproduce the experimental results. The appendices contain the proofs for all the theoretical results in this paper. We also made sure it is clear what assumptions are made in each theorem, proposition, lemma, and corollary.

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

## THE USE OF LARGE LANGUAGE MODELS

We used OpenAI's ChatGPT and Codex for the following assistance purposes:

- Detect grammatical errors and improve clarity.
- Assist in writing the code for generating the figures in the paper.

## A   RELATED WORK

Estimation of the Bayes error $\mathrm{Err}^*$ (see Section 2.1 for the definition) is a classical problem in the field of machine learning and pattern recognition. Existing methods for Bayes error estimation can be categorized based on the type of data used, specifically as either instance-label pair-based or soft label-based estimation.

Most of the existing methods require a dataset $\{(x_i, y_i)\}_{i=1}^n$ composed of instances $x_i \in \mathcal{X}$ paired with their respective labels $y_i \in \mathcal{Y} = \{0, 1\}$. Assuming the two class-conditional distributions of instances have densities satisfying certain conditions, Berisha et al. (2014), Moon et al. (2018) and Noshad et al. (2019) proposed approaches based on the estimation of $f$-divergence between the class-conditional densities. Specifically, Berisha et al. (2014) and Moon et al. (2018) suggested estimating upper or lower bounds and thus their methods suffer from relatively large biases. Although Noshad et al. (2019) succeeded in estimating the exact Bayes error instead of bounds on it, they assumed that the class-conditional densities $p_0, p_1$ are Hölder-continuous and satisfy $0 < L \le p_i \le U$ $(i = 0, 1)$ for some constants $L$ and $U$, and their estimator requires the knowledge of the values of these constants, which can be unpractical.

On the other hand, Theisen et al. (2021) proposed a Bayes error estimation method based on normalizing flow models (Papamakarios et al., 2021; Kingma and Dhariwal, 2018). They first showed that the Bayes error is invariant under invertible transformations. Using this result, they suggested approximating the data distribution with a normalizing flow and then computing the Bayes error for its base Gaussian distribution, which can be done using the Holmes–Diaconis–Ross integration scheme (Diaconis and Holmes, 1995; Gessner et al., 2020). One of the drawbacks of their approach is that it is prohibitively memory-intensive for high-dimensional data, as mentioned in their paper.

The method proposed by Ishida et al. (2023), described in Section 2.1, was unique in that it utilized the soft labels $\eta_i = \eta(x_i)$ instead of the instances $x_i$ themselves. Jeong et al. (2023) extended the approach of Ishida et al. (2023) to the estimation of the Bayes error in multi-class classification problems.

## B   SUPPLEMENTARY FOR SECTION 2

### B.1   PROOF OF THEOREM 1

For each $i = 1, \ldots, d$, let $\widehat{p}_i = \frac{1}{m} \sum_{j=1}^m Z_i^{(j)}$ where $Z_i^{(1)}, \ldots, Z_i^{(m)} \in \{0, 1\}$ are independent Bernoulli random variables with mean $p_i$. For ease of notation, we denote $\widehat{p} = (\widehat{p}_1, \ldots, \widehat{p}_d)$ and $p = (p_1, \ldots, p_d)$. Noting that $\mathbb{E}[\widehat{p}] = p$, $\widehat{p}$ is an unbiased and consistent estimator of $p$.

Suppose that we would like to estimate the value $\phi(p)$ for some function $\phi : \mathbb{R}^d \to \mathbb{R}$. It is natural to consider a plug-in estimator $\phi(\widehat{p})$. We can evaluate its bias $\mathbb{E}[\phi(\widehat{p})] - \phi(p) = \mathbb{E}[\phi(\widehat{p})] - \phi(\mathbb{E}[\widehat{p}])$ using a sharpened version of Jensen's inequality by Gao et al. (2019).

**Lemma 1.** *Suppose $\phi$ is differentiable at $\mu$. Then, we have*

$$\frac{\inf_{z \neq \mu} h(z)}{m} \sum_{i=1}^d p_i(1 - p_i) \le \mathbb{E}[\phi(\widehat{p})] - \phi(p) \le \frac{\sup_{z \neq \mu} h(z)}{m} \sum_{i=1}^d p_i(1 - p_i), \qquad (8)$$

*where*

$$h(z) := \frac{\phi(z) - \phi(\mu) - \nabla\phi(\mu)^\top (z - \mu)}{\|z - \mu\|_2^2}. \qquad (9)$$

*Proof.* For any $i, j \in [d]$, we have

$$\mathbb{E}\left[(\widehat{p}_i - p_i)(\widehat{p}_j - p_j)\right] = \frac{1}{m}p_i(1 - p_i)\delta_{ij}, \tag{10}$$

where $\delta_{ij} = \mathbb{1}\left[i = j\right]$ is the Kronecker delta. This implies

$$\mathrm{tr}\left(\mathrm{Cov}\left[\widehat{p}\right]\right) = \frac{1}{m}\sum_{i=1}^{d} p_i(1 - p_i). \tag{11}$$

Now, we can apply (2.4) in Gao et al. (2019) to conclude the proof.[7] $\quad\square$

It is often the case that the function $\phi$ in question is Lipschitz continuous. For example, every convex function is locally Lipschitz on any convex compact subset of the relative interior of its domain (see e.g. Hiriart-Urruty and Lemaréchal, 1993, Theorem 3.1.2 in Chapter IV). In such cases, we can derive another type of bound based on Lipschitzness.

**Lemma 2** (Lipschitzness-based bounds for the bias)**.** *If $\phi$ is L-Lipschitz with respect to the 1-norm, we have*

$$\left|\mathbb{E}\left[\phi(\widehat{p})\right] - \phi(p)\right| \leq Ld\sqrt{\frac{\pi}{2m}}. \tag{12}$$

*Proof.* First of all, it holds that

$$\left|\mathbb{E}\left[\phi(\widehat{p})\right] - \phi(p)\right| \leq \mathbb{E}\left[|\phi(\widehat{p}) - \phi(p)|\right] \tag{13}$$

$$\leq L\,\mathbb{E}\left[\sum_{i=1}^{d} |\widehat{p}_i - p_i|\right] \quad \text{(by Lipschitzness)} \tag{14}$$

$$= L\sum_{i=1}^{d} \mathbb{E}\left[|\widehat{p}_i - p_i|\right]. \tag{15}$$

Each summand can be bounded by integrating Hoeffding's tail bound (see, e.g., Theorem 2.2.6 of (Vershynin, 2018)):

$$\mathbb{E}\left[|\widehat{p}_i - p_i|\right] = \int_0^\infty \mathbb{P}\left(|\widehat{p}_i - p_i| \geq t\right)\,dt \tag{16}$$

$$\leq \int_0^\infty 2\exp(-2mt^2)\,dt \tag{17}$$

$$= \sqrt{\frac{\pi}{2m}}. \tag{18}$$

Hence the result follows. $\quad\square$

Lemma 2 suggests that the bias is at most $\mathcal{O}(1/\sqrt{m})$, whereas Lemma 1 indicates a faster convergence rate of $\mathcal{O}(1/m)$ whenever the supremum and infimum are finite.

Now, define

$$\phi_{\mathrm{Err}}(z) := \min\left\{z, 1 - z\right\} \tag{19}$$

for $z \in [0, 1]$. We choose $\phi = \phi_{\mathrm{Err}}$ and apply Lemma 1 and Lemma 2 to show the following lemma.

**Lemma 3.** *For each $i \in [d]$, we have*

$$-\sqrt{\frac{\pi}{2m}} \leq \mathbb{E}\left[\phi_{\mathrm{Err}}(\widehat{p}_i)\right] - \phi_{\mathrm{Err}}(p_i) \leq 0. \tag{20}$$

*Furthermore, if $p_i \neq 0.5$, it holds that*

$$-\frac{p_i(1 - p_i)}{m|2p_i - 1|} \leq \mathbb{E}\left[\phi_{\mathrm{Err}}(\widehat{p}_i)\right] - \phi_{\mathrm{Err}}(p_i) \leq 0. \tag{21}$$

---

[7]Although Gao et al. (2019) only discusses the univariate case, it is straightforward to extend the result to the multivariate case.

*Proof.* The lower bound of the first inequality is a direct consequences of Lemma 2 and the fact that $\phi_{\text{Err}}$ is 1-Lipschitz. The upper bound follows from the concavity of $\phi_{\text{Err}}$ and the classic Jensen inequality.

To prove the second inequality, we first assume $0 \le p_i < 0.5$. For any $z \in [0,1] \setminus \{p_i\}$, we have

$$h(z) = \frac{\phi_{\text{Err}}(z) - \phi_{\text{Err}}(p_i) - (z - p_i)}{(z - p_i)^2} = \begin{cases} 0 & \text{for } z \in [0, 0.5], \\ \dfrac{1 - 2z}{(z - p_i)^2} & \text{for } z \in [0.5, 1], \end{cases} \tag{22}$$

and thus

$$-\frac{1}{1 - 2p_i} \le h(z) \le 0. \tag{23}$$

Therefore, Lemma 1 implies

$$-\frac{p_i(1 - p_i)}{m(1 - 2p_i)} \le \mathbb{E}\left[\phi_{\text{Err}}(\widehat{p}_i)\right] - \phi_{\text{Err}}(p_i) \le 0. \tag{24}$$

Next, we assume $0.5 < p_i \le 1$. A similar argument proves $-\frac{1}{2p_i - 1} \le h(z) \le 0$ and

$$-\frac{p_i(1 - p_i)}{m(2p_i - 1)} \le \mathbb{E}\left[\phi_{\text{Err}}(\widehat{p}_i)\right] - \phi_{\text{Err}}(p_i) \le 0. \tag{25}$$

Combining (24) and (25) proves the second bound. $\qquad\square$

Note that Lemma 3 can be rewritten as

$$-\min\left\{\frac{L_{\text{Err}}(p_i)}{m}, \sqrt{\frac{\pi}{2m}}\right\} \le \mathbb{E}\left[\phi_{\text{Err}}(\widehat{p}_i)\right] - \phi_{\text{Err}}(p_i) \le 0, \tag{26}$$

where

$$L_{\text{Err}}(q) = \begin{cases} \dfrac{q(1 - q)}{|2q - 1|} & \text{if } q \ne 0.5, \\ +\infty & \text{if } q = 0.5. \end{cases} \tag{27}$$

Fig. 5 shows the graph of the function $L_{\text{Err}}$. Finally, Theorem 1 is proved as follows.

*Proof of Theorem 1.* Conditioning on $x$, let $\{y^{(j)}\}_{j=1}^m$ be independent Bernoulli random variables with mean $\eta(x)$ and $\widehat{\eta}$ be their average $\frac{1}{m}\sum_{j=1}^m y^{(j)}$. Then, Lemma 3 gives

$$-\min\left\{\frac{L_{\text{Err}}(\eta(x))}{m}, \sqrt{\frac{\pi}{2m}}\right\} \le \mathbb{E}\left[\phi_{\text{Err}}(\widehat{\eta}) \mid x\right] - \phi_{\text{Err}}(\eta(x)) \le 0. \tag{28}$$

By taking expectation over $x$, we obtain

$$-\mathbb{E}\left[\min\left\{\frac{L_{\text{Err}}(\eta(x))}{m}, \sqrt{\frac{\pi}{2m}}\right\}\right] \le \mathbb{E}\left[\phi_{\text{Err}}(\widehat{\eta})\right] - \mathbb{E}\left[\phi_{\text{Err}}(\eta(x))\right] \le 0. \tag{29}$$

Now the claim follows since $\mathbb{E}\left[\widehat{\text{Err}}^*(\widehat{\eta}_{1:n})\right] = \mathbb{E}\left[\phi_{\text{Err}}(\widehat{\eta})\right]$ and $\text{Err}^* = \mathbb{E}\left[\phi_{\text{Err}}(\eta(x))\right]$. $\qquad\square$

## B.2 PROOFS FOR OTHER RESULTS

### B.2.1 SUPERIORITY OF THEOREM 1 OVER THE EXISTING RESULT

**Proposition 1.** *Theorem 1 is tighter than the existing result* (2) *by Ishida et al.* (2023) *for all* $n, m \ge 1$.

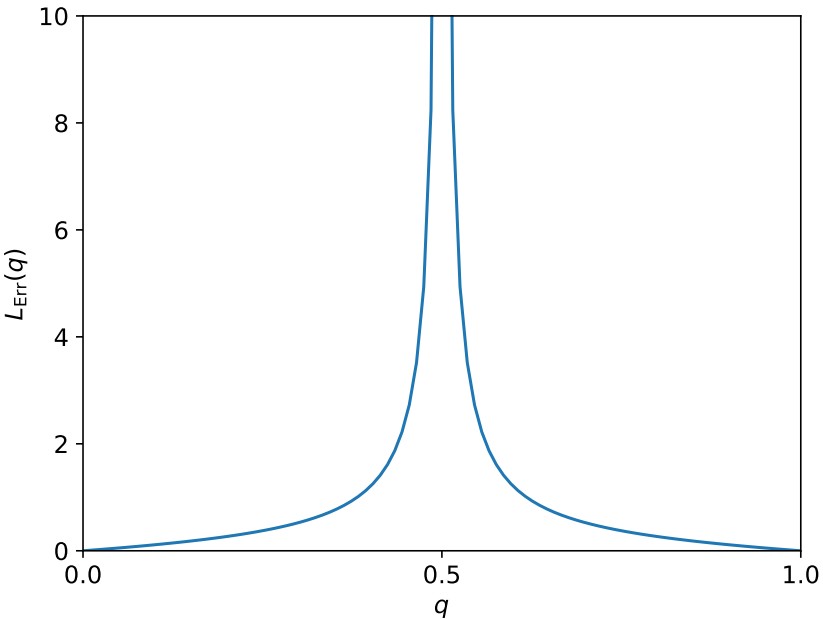

Figure 5: The graph of the function $L_{\mathrm{Err}}$.

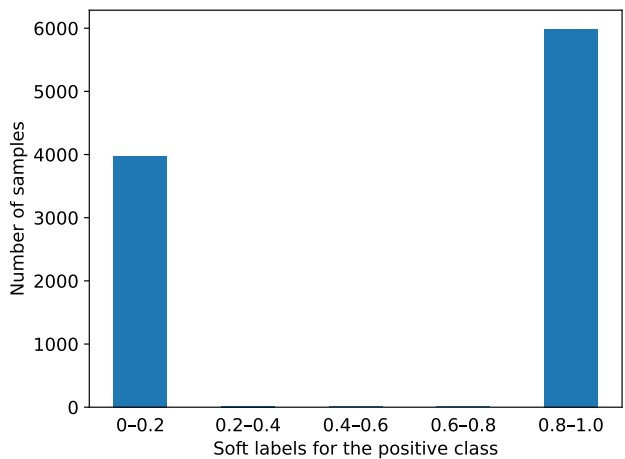

Figure 6: The distribution of the soft labels for the positive class in the binarized CIFAR-10H dataset (see Section 4.1 for details). You can see the two classes are very well-separated.

*Proof.* Our upper bound $0$ is of course smaller than the upper bound in (2). As for the lower bounds, we can see that the condition

$$\sqrt{\frac{\pi}{2m}} \le \frac{1}{2\sqrt{m}} + \sqrt{\frac{\log(2n\sqrt{m})}{m}} \tag{30}$$

is equivalent to

$$n\sqrt{m} \ge \frac{1}{2} \exp\left(\left(\frac{\sqrt{2\pi}-1}{2}\right)^2\right). \tag{31}$$

The right-hand side is less than $1$ so (31) always holds. $\qquad\square$

### B.2.2 Faster decay rate of the bias in well-separated cases (Corollary 1 & Example 1)

*Proof of Corollary 1.* By the assumption, we have

$$L_{\mathrm{Err}}(\eta(x)) = \frac{\eta(x)(1 - \eta(x))}{2\left|\eta(x) - 0.5\right|} \leq \frac{1/4 - c^2}{2c} \tag{32}$$

almost surely. Combining this with Theorem 1, we obtain the result. □

*Proof of Example 1.* Let $f_0, f_1$ be the densities of $\mathcal{F}_0, \mathcal{F}_1$, respectively. From the data generation model, we can see that

$$\eta(x) = \frac{\nu f_0(x) + (1 - \nu) f_1(x)}{f_0(x) + f_1(x)} \tag{33}$$

for any $x \in \mathcal{X}$ such that $f_0(x) > 0$ or $f_1(x) > 0$. Since the supports of $\mathcal{F}_0, \mathcal{F}_1$ are disjoint, we can assume that

$$f_0(x) > 0 \implies f_1(x) = 0, \quad f_1(x) > 0 \implies f_0(x) = 0 \quad \text{for any } x \in \mathcal{X}, \tag{34}$$

which implies

$$\eta(x) = \begin{cases} \nu & \text{if } f_0(x) > 0, \\ 1 - \nu & \text{if } f_1(x) > 0. \end{cases} \tag{35}$$

Therefore, it holds that

$$|\eta(x) - 0.5| = |\nu - 0.5| = c > 0 \tag{36}$$

almost surely. □

### B.2.3 Computable bound of the bias (Corollary 2)

**Lemma 4.** *If* $\mathrm{Err}^* \leq E$*, we have*

$$\mathbb{P}\left(\phi_{\mathrm{Err}}(\eta(x)) \geq t\right) \leq \min\left\{1, \frac{E}{t}\right\} \tag{37}$$

*for any* $t \in (0, 1/2]$.

*Proof.* Since $\phi_{\mathrm{Err}}(\eta(x))$ is a non-negative random variable with mean $\mathbb{E}\left[\phi_{\mathrm{Err}}(\eta(x))\right] = \mathrm{Err}^*$, Markov's inequality gives

$$\mathbb{P}\left(\phi_{\mathrm{Err}}(\eta(x)) \geq t\right) \leq \frac{\mathrm{Err}^*}{t} \leq \frac{E}{t} \tag{38}$$

for any $t > 0$. □

*Proof of Corollary 2.* Let $z = \phi_{\mathrm{Err}}(\eta(x))$. Since

$$\frac{\eta(x)(1 - \eta(x))}{|2\eta(x) - 1|} = \frac{z(1 - z)}{1 - 2z}, \tag{39}$$

we have

$$\frac{\eta(x)(1 - \eta(x))}{|2\eta(x) - 1|} < \frac{t(1 - t)}{1 - 2t} \tag{40}$$

if $z < t$. Therefore,

$$\mathbb{E}\left[\min\left\{\frac{L_{\mathrm{Err}}(\eta(x))}{m}, \sqrt{\frac{\pi}{2m}}\right\}\right] \tag{41}$$

$$\leq \mathbb{E}\left[\sqrt{\frac{\pi}{2m}} \cdot \mathbb{1}\left[z \geq t\right] + \frac{1}{m} \cdot \frac{t(1 - t)}{1 - 2t} \cdot \mathbb{1}\left[z < t\right]\right] \tag{42}$$

$$\leq \sqrt{\frac{\pi}{2m}} \cdot \mathbb{P}\left(z \geq t\right) + \frac{1}{m} \cdot \frac{t(1 - t)}{1 - 2t} \cdot 1. \tag{43}$$

By using Lemma 4, we obtain

$$\mathbb{E}\left[\min\left\{\frac{L_{\mathrm{Err}}(\eta(x))}{m}, \sqrt{\frac{\pi}{2m}}\right\}\right] \leq \sqrt{\frac{\pi}{2m}} \cdot \min\left\{1, \frac{E}{t}\right\} + \frac{1}{m} \cdot \frac{t(1 - t)}{1 - 2t}. \tag{44}$$

Combining this with Theorem 1 yields the result. □

### B.2.4 STATISTICAL CONSISTENCY

**Corollary 3.**    *(i) For any $\delta \in (0, 1)$, with probability at least $1 - \delta$, we have*

$$\left|\widehat{\mathrm{Err}^*}(\widehat{\eta}_{1:n}) - \mathrm{Err}^*\right| \leq \sqrt{\frac{\log(2/\delta)}{2n}} + \sqrt{\frac{\pi}{2m}}. \tag{45}$$

*(ii) Suppose there exists a constant $c > 0$ such that $|\eta(x) - 0.5| \geq c$ holds almost surely. Then, for any $\delta \in (0, 1)$, with probability at least $1 - \delta$, we have*

$$\left|\widehat{\mathrm{Err}^*}(\widehat{\eta}_{1:n}) - \mathrm{Err}^*\right| \leq \sqrt{\frac{\log(2/\delta)}{2n}} + \frac{1 - 4c^2}{8cm}. \tag{46}$$

*Proof.* By Hoeffding's inequality, we have

$$\left|\widehat{\mathrm{Err}^*}(\widehat{\eta}_{1:n}) - \mathbb{E}\left[\widehat{\mathrm{Err}^*}(\widehat{\eta}_{1:n})\right]\right| \tag{47}$$

$$\leq \left|\widehat{\mathrm{Err}^*}(\widehat{\eta}_{1:n}) - \mathbb{E}\left[\widehat{\mathrm{Err}^*}(\widehat{\eta}_{1:n})\right]\right| + \left|\mathbb{E}\left[\widehat{\mathrm{Err}^*}(\widehat{\eta}_{1:n})\right] - \mathrm{Err}^*\right| \tag{48}$$

$$\leq \sqrt{\frac{\log(2/\delta)}{2n}} + \left|\mathbb{E}\left[\widehat{\mathrm{Err}^*}(\widehat{\eta}_{1:n})\right] - \mathrm{Err}^*\right| \tag{49}$$

with probability greater than $1 - \delta$. By upper-bounding the second term using Theorem 1 and Corollary 1, we obtain (45) and (46), respectively. □

### B.3 NUMERICAL EXPERIMENTS

Here, we examine the validity of our theory using synthetic datasets composed of instances drawn from the following distributions.[8]

(a) The Gaussian mixture $\mathbb{P}_{\mathcal{X}} = 0.5 \cdot \mathbb{P}_0 + 0.5 \cdot \mathbb{P}_1$ with $\mathbb{P}_0 = N((0, 0), I_2)$ and $\mathbb{P}_1 = N((2, 2), I_2)$.

(b) The Gaussian mixture $\mathbb{P}_{\mathcal{X}} = 0.5 \cdot \mathbb{P}_0 + 0.5 \cdot \mathbb{P}_1$ with the completely overlapping components $\mathbb{P}_0 = \mathbb{P}_1 = N((0, 0), I_2)$.

(c) The distribution with label flips discussed in Example 1. We set the label flip rate to $\nu = 0.1$ and use the uniform distributions over $[0, 1)^2$ and $[1, 2)^2$ as $\mathcal{F}_0$ and $\mathcal{F}_1$, respectively. [9]

Note that the "perfect separation" assumption of Corollary 1 is met only by **(c)**. For each $m = 10, 25, 50, 100, 250, 500, 1000$, we perform the following procedure 1000 times:

(i) Sample $n = 2000$ instances from one of the distributions **(a)**, **(b)** and **(c)**.

(ii) For each instance $x_i$, generate $m$ hard labels $y_i^{(1)}, \ldots, y_i^{(m)}$ from the posterior class distribution $\mathbb{P}(y \mid x = x_i)$ and compute the approximate soft label $\widehat{\eta}_i = \frac{1}{m} \sum_{j=1}^m y_i^{(j)}$.

(iii) Compute the estimate $\widehat{\mathrm{Err}^*}(\widehat{\eta}_{1:n})$.

Then, we approximate the expectation $\mathbb{E}\left[\widehat{\mathrm{Err}^*}(\widehat{\eta}_{1:n})\right]$ by the average of the 1000 estimates to calculate the bias $\left|\mathbb{E}\left[\widehat{\mathrm{Err}^*}(\widehat{\eta}_{1:n})\right] - \mathrm{Err}^*\right|$.

Fig. 7 is a log-log plot showing the empirical bias (the blue solid line) as a function of $m$ for each setup. The corresponding theoretical bound (3) is also shown by the black dashed line.[10] We note

---

[8]This experiment takes around 1 hour for each distribution on a CPU.

[9]Note that the choice of base distributions $\mathcal{F}_0, \mathcal{F}_1$ does not matter as long as they satisfy the assumption (34) because $\eta$ is determined solely by the label flip rate $\nu$; see (35).

[10]The expectation $\mathbb{E}\left[\min\left\{\frac{L_{\mathrm{Err}}(\eta(x))}{m}, \sqrt{\frac{\pi}{2m}}\right\}\right]$ is approximated by the sample average over 20000 data points.

that the empirically observed bias is smaller than the theoretical bound in all the setups as expected. Our theory accurately predicts the decay of the bias, especially in **(a)** and **(b)**. If we fit a function of the form $m^p$ to a bias curve, the slope of its graph corresponds to the exponent $p$. The slopes obtained by least-squares fitting are $-0.9066$ for **(a)**, $-0.4970$ for **(b)**, and $-0.4228$ for **(c)**. Recall that, the two class-conditional distributions were completely overlapping with each other in **(b)**. Thus the slope close to $-0.5$ is as expected. What is somewhat interesting is the result for **(a)**. Although this setup does not satisfy the perfect separation assumption of Corollary 1, the observed bias decay is approximately proportional to $m^{-1}$. It suggests that the "fast" $m^{-1}$ term dominates the "slow" $m^{-1/2}$ term. As for **(c)**, examining the slope $-0.4228$ will not make much sense as the shape of the graph Fig. 7c is far from being a straight line.

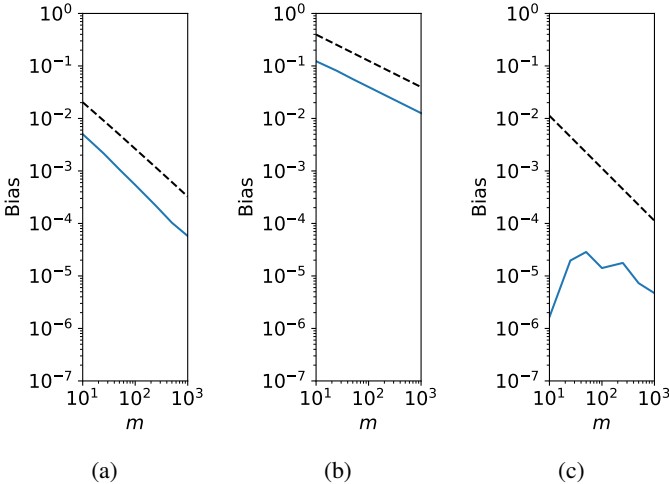

(a)  (b)  (c)

Figure 7: The bias of the hard-label-based estimator $\widehat{\mathrm{Err}}^*(\widehat{\eta}_{1:n})$ as a function of the number $m$ of hard labels per sample. The blue solid lines indicate the experimental results while the black dashed lines indicate the theoretical upper bounds in Theorem 1.

## C   SUPPLEMENTARY FOR SECTION 3

In this section, we present the proof of Theorem 2. Appendix C.1 presents a new risk bound for binary isotonic regression (Proposition 2) as well as a useful lemma for general shape-constrained nonparametric regression problems (Lemma 5). Then, we employ these results to prove the theorem in Appendix C.2.

### C.1   RISK BOUND FOR BINARY ISOTONIC REGRESSION

#### C.1.1   NONPARAMETRIC REGRESSION AND ISOTONIC REGRESSION

Here, we introduce general nonparametric regression problems where we aim to estimate the underlying signal from noisy observations. Then, we describe the isotonic regression setting.

Let $T$ be a set. Assume that, for each design point $t_i \in T$, $i = 1, \ldots, n$, we observe

$$y_i = f^*(t_i) + \xi_i, \tag{50}$$

where $f^* : T \to \mathbb{R}$ is the unknown regression function and $\xi_i \in \mathbb{R}$ are independent and mean-zero noise variables. A natural estimator would be the least squares estimator (LSE)

$$\widehat{f} \in \arg\min_{f \in \mathcal{F}} \frac{1}{n} \sum_{i=1}^{n} (y_i - f(t_i))^2, \tag{51}$$

where $\mathcal{F}$ is some pre-defined function class. In the fixed design setting, the quality of an estimator $\widehat{f}$ is evaluated by the $\ell^2$ risk

$$\frac{1}{n} \sum_{i=1}^{n} \left( \widehat{f}(t_i) - f^*(t_i) \right)^2. \tag{52}$$

Under this criterion, estimators are evaluated only on the fixed design points $t_i$ so estimating the function $f^*$ is equivalent to estimating the sequence/vector $\boldsymbol{\mu} := (f^*(t_1), \ldots, f^*(t_n)) \in \mathbb{R}^n$.

From this perspective, the regression problem can be reformulated as follows. Our observation is an $n$-dimensional random vector $\boldsymbol{y} = (y_1, \ldots, y_n) \in \mathbb{R}^n$ of the form

$$\boldsymbol{y} = \boldsymbol{\mu} + \boldsymbol{\xi}. \tag{53}$$

Here $\boldsymbol{\mu} = (\mu_1, \ldots, \mu_n) \in \mathbb{R}^n$ is the unknown signal and $\boldsymbol{\xi} = (\xi_1, \ldots, \xi_n) \in \mathbb{R}^n$ is a centered noise vector whose elements are independent. Let $K \subset \mathbb{R}^n$ be a closed convex subset from which we choose our estimates, which corresponds to the function class $\mathcal{F}$ in the function estimation formulation described above. Often it is assumed that the true signal $\boldsymbol{\mu}$ indeed belongs to $K$. However, in our results in Appendix C.1, we allow model misspecification, i.e., we do not assume $\boldsymbol{\mu} \in K$. The LSE $\widehat{\boldsymbol{\mu}}$ is the Euclidean projection of $\boldsymbol{y}$ onto $K$:

$$\widehat{\boldsymbol{\mu}} = \arg\min_{\boldsymbol{u} \in K} \|\boldsymbol{y} - \boldsymbol{u}\|_2^2. \tag{54}$$

**Isotonic regression**  *Isotonic regression* is a special case where we choose $K$ to be the collection $\mathcal{M}_n$ of all non-decreasing sequences of length $n$:

$$\mathcal{M}_n := \left\{ \boldsymbol{u} \in \mathbb{R}^n \mid u_1 \leq \ldots \leq u_n \right\}. \tag{55}$$

Note that $\mathcal{M}_n$ is a closed convex cone. Here the goal is to estimate isotonic, or monotonic, signals from noisy observations. Recall that the LSE $\widehat{\boldsymbol{\mu}}$ is the Euclidean projection of the observation vector $\boldsymbol{y}$ onto $\mathcal{M}_n$:

$$\widehat{\boldsymbol{\mu}} = \arg\min_{\boldsymbol{u} \in \mathcal{M}_n} \|\boldsymbol{y} - \boldsymbol{u}\|_2^2. \tag{56}$$

It has the following explicit representation (Robertson et al., 1988), which is known as the min-max formula:

$$\widehat{\mu}_i = \min_{l \geq i} \max_{k \leq i} \bar{y}_{k:l}, \quad i = 1, \ldots, n, \tag{57}$$

where $\bar{y}_{k:l} := \frac{1}{l-k+1} \sum_{i=k}^{l} y_i$ is the average of $y_k, \ldots, y_l$. It can be efficiently computed with the *pool adjacent violators* (PAV) algorithm (Ayer et al., 1955).

We are interested in evaluating the risk $\frac{1}{n}\|\widehat{\boldsymbol{\mu}} - \boldsymbol{\mu}\|_2^2$ of the LSE $\widehat{\boldsymbol{\mu}}$, which has been extensively studied in the literature (e.g. Zhang, 2002; Chatterjee, 2014; Chatterjee et al., 2015; Bellec and Tsybakov, 2015; Bellec, 2018; Yang and Barber, 2019; Chatterjee and Lafferty, 2019). We will cover the results from these existing works later in Appendix C.1.2.

### C.1.2  REVIEW OF THE EXISTING RISK BOUNDS FOR ISOTONIC REGRESSION

First, we introduce some notions that will be needed below. For each non-decreasing sequence $\boldsymbol{u} \in \mathcal{M}_n$, we denote its total variation by

$$V(\boldsymbol{u}) := \max_i u_i - \min_i u_i. \tag{58}$$

We also let $k(\boldsymbol{u})$ be the number of constant pieces in $\boldsymbol{u}$. In other words, $k(\boldsymbol{u}) - 1$ is the number of the inequalities $u_i \leq u_{i+1}$ that are strict, so the sequence $u_1, \ldots, u_n$ has $k(\boldsymbol{u}) - 1$ jumps in total.

For the cases where $\boldsymbol{\mu} \in \mathcal{M}_n$ and the noises have bounded variance $\mathbb{E}\left[\xi_i^2\right] \leq \sigma^2$, the following bound on the expected risk was proven by Zhang (2002):

$$\mathbb{E}\left[\frac{1}{n}\|\widehat{\boldsymbol{\mu}} - \boldsymbol{\mu}\|_2^2\right] \leq C \left\{ \left( \frac{\sigma^2 V(\boldsymbol{\mu})}{n} \right)^{2/3} + \frac{\sigma^2 \log(en)}{n} \right\}, \tag{59}$$

where $C \leq 12.3$ is an absolute constant. Chatterjee et al. (2015) showed this $n^{-2/3}$ rate is minimax, while providing another type of risk bound

$$\mathbb{E}\left[\frac{1}{n}\|\widehat{\boldsymbol{\mu}} - \boldsymbol{\mu}\|_2^2\right] \leq 6 \min_{\boldsymbol{u} \in \mathcal{M}_n} \left\{ \frac{1}{n}\|\boldsymbol{u} - \boldsymbol{\mu}\|_2^2 + \frac{\sigma^2 k(\boldsymbol{u})}{n} \log \frac{en}{k(\boldsymbol{u})} \right\} \tag{60}$$

under the assumptions that $\boldsymbol{\mu} \in \mathcal{M}_n$ and $\xi_i$ are i.i.d. with finite variance $\mathbb{E}\left[\xi_i^2\right] = \sigma^2$. (60) is adaptive, unlike (59), in the sense that it gives a parametric rate $n^{-1/2}$ up to logarithmic factors when the true signal $\boldsymbol{\mu}$ is well-approximated by some $\boldsymbol{u} \in \mathcal{M}_n$ with small $k(\boldsymbol{u})$ (i.e., a piecewise constant sequence with not too many pieces). Later, Bellec (2018) showed two types of bounds, improving the previous results in the case of i.i.d. Gaussian noise $\boldsymbol{\xi} \sim N(0, \sigma^2 I_n)$. In the first result, they proved that, with probability greater than $1 - e^{-x}$, we have

$$\frac{1}{n}\|\widehat{\boldsymbol{\mu}} - \boldsymbol{\mu}\|_2^2 \leq \min_{\boldsymbol{u} \in \mathcal{M}_n}\left\{\frac{1}{n}\|\boldsymbol{u} - \boldsymbol{\mu}\|_2^2 + 2c\sigma^2\left(\frac{\sigma + V(\boldsymbol{u})}{\sigma n}\right)^{2/3}\right\} + \frac{4\sigma^2 x}{n}, \tag{61}$$

where $c$ is an absolute constant. A corresponding bound in expectation also can be derived by integrating this high-probability bound. The second result is that, with probability at least $1 - e^{-x}$, we have

$$\frac{1}{n}\|\widehat{\boldsymbol{\mu}} - \boldsymbol{\mu}\|_2^2 \leq \min_{\boldsymbol{u} \in \mathcal{M}_n}\left\{\frac{1}{n}\|\boldsymbol{u} - \boldsymbol{\mu}\|_2^2 + \frac{2\sigma^2 k(\boldsymbol{u})}{n}\log\frac{en}{k(\boldsymbol{u})}\right\} + \frac{4\sigma^2 x}{n}. \tag{62}$$

A similar in-expectation bound

$$\mathbb{E}\left[\frac{1}{n}\|\widehat{\boldsymbol{\mu}} - \boldsymbol{\mu}\|_2^2\right] \leq \min_{\boldsymbol{u} \in \mathcal{M}_n}\left\{\frac{1}{n}\|\boldsymbol{u} - \boldsymbol{\mu}\|_2^2 + \frac{\sigma^2 k(\boldsymbol{u})}{n}\log\frac{en}{k(\boldsymbol{u})}\right\} \tag{63}$$

also holds. Bellec (2018)'s results, (61), (62) and (63), have several features worth mentioning. First, their leading constants are 1. For this reason, these bounds are called *sharp* oracle inequalities. Second, they are valid even under model misspecification, which (59) nor (60) allowed. Third, (61) and (62) were the first oracle inequalities that were shown to hold with high probability, rather than in expectation. The last point is especially important for our purpose, i.e., computing confidence intervals. A major drawback of the results by Bellec (2018) is that they are restricted to Gaussian noise. Yang and Barber (2019) employed their unique sliding window norm technique to prove the following bound for general sub-Gaussian noise with variance proxy $\sigma^2$:

$$\mathbb{E}\left[\frac{1}{n}\|\widehat{\boldsymbol{\mu}} - \boldsymbol{\mu}\|_2^2\right] \leq 48\left(\frac{\sigma^2 V(\boldsymbol{\mu})\log(2n)}{n}\right)^{2/3} + \frac{96\sigma^2 \log^2(2n)}{n} \tag{64}$$

Under model misspecification $\boldsymbol{\mu} \notin \mathcal{M}_n$, (64) still remains valid with $\boldsymbol{\mu}$ replaced by its projection onto $\mathcal{M}_n$. A similar high-probability bound also can be derived by almost the same argument, although they did not mention it in their paper.

### C.1.3 METRIC ENTROPY BOUNDS FOR ISOTONIC CONSTRAINTS

For real numbers $-\infty < a < b < \infty$, we define the truncated version of the isotonic cone $\mathcal{M}_n$ as

$$\mathcal{M}_n(a, b) := \{\boldsymbol{x} \in \mathbb{R}^n \mid a \leq x_1 \leq \ldots \leq x_n \leq b\}. \tag{65}$$

$\mathcal{M}_n(a, b)$ is not a cone, unlike $\mathcal{M}_n$, but it is still a closed convex set. We also define the set of all non-decreasing functions from $[0, 1)$ to $[0, 1]$:

$$\mathcal{M} := \{f : [0, 1) \to [0, 1] \mid f \text{ is non-decreasing}\}. \tag{66}$$

Let $(\mathcal{F}, \|\cdot\|)$ be a subset of a normed function space. Given two functions $l, u \in \mathcal{F}$ with $\|u - l\| \leq \varepsilon$, the set

$$\{f \in \mathcal{F} \mid l \leq f \leq u\} \tag{67}$$

is called an $\varepsilon$-*bracket* (Van Der Vaart and Wellner, 1996). The $\varepsilon$-*bracketing number* $\mathcal{N}_{[]}(\mathcal{F}, \|\cdot\|, \varepsilon)$ of $(\mathcal{F}, \|\cdot\|)$ is the smallest number of $\varepsilon$-brackets needed to cover $\mathcal{F}$. The logarithm of bracketing numbers is called *bracketing entropy*.

Van Der Vaart and Wellner (1996, Theorem 2.7.5) and Gao and Wellner (2007, Theorem 1.1) proved the $\varepsilon$-bracketing entropy of $\mathcal{M}$ is of order $\varepsilon^{-1}$, i.e.,

$$\log \mathcal{N}_{[]}(\mathcal{M}, \|\cdot\|_{L^p}, \varepsilon) \leq \frac{C_p}{\varepsilon}, \quad \forall \varepsilon > 0, \tag{68}$$

where $C_p > 0$ is a universal constant depending only on $p \in [1, \infty)$ and $\|\cdot\|_{L^p}$ is the $L^p$ norm under Lebesgue measure. Later, Chatterjee (2014, Lemma 4.20) established a tool that enables us to

convert the bracketing entropy bound (68) for monotone functions into a metric entropy bound for monotone sequences. It has been commonly utilized in previous studies (Chatterjee, 2014; Bellec, 2018; Chatterjee and Lafferty, 2019). Although the original result by Chatterjee (2014) was stated for the Euclidean norm $\|\cdot\|_2$, results for other $p$-norms $\|\cdot\|_p$, $p \in [1, \infty]$ can be obtained by a similar argument. We state and prove this generalized version below.

**Theorem 4.** *Take any $p \in [1, \infty)$. Then, we have*

$$\log \mathcal{N}(\mathcal{M}_n(a, b), \|\cdot\|_p, \varepsilon) \leq \frac{C_p(b - a)n^{1/p}}{\varepsilon} \tag{69}$$

*for $\varepsilon > 0$. Here $C_p$ is the same constant as in (68).*

*Proof.* Without loss of generality, we assume $a = 0$ and $b = 1$. First of all, note the general fact that $\varepsilon$-covering number is upper-bounded by $2\varepsilon$-bracketing number (see, e.g. Van Der Vaart and Wellner, 1996). This, together with the bracketing number bound (68), implies

$$\log \mathcal{N}(\mathcal{M}, \|\cdot\|_{L^p}, \varepsilon) \leq \log \mathcal{N}_{[]}(\mathcal{M}, \|\cdot\|_{L^p}, 2\varepsilon) \leq \frac{C_p}{2\varepsilon}. \tag{70}$$

Therefore, there exists an $\varepsilon$-net $\tilde{N}$ of the function class $\mathcal{M}$ with $\log|\tilde{N}| \leq \frac{C_p}{2\varepsilon}$. Now set $\delta = 2n^{1/p}\varepsilon$. We will construct a $\delta$-net $N$ of the sequence class $\mathcal{M}_n(0, 1)$ based on $\tilde{N}$. To this end, for each monotone sequence $\boldsymbol{u} \in \mathcal{M}_n(0, 1)$, we associate it with a monotone piecewise constant function $g_{\boldsymbol{u}} \in \mathcal{M}$ of the form

$$g_{\boldsymbol{u}}(x) = \sum_{i=1}^{n} u_i \mathbb{1}\left[x \in \left[\frac{i-1}{n}, \frac{i}{n}\right)\right]. \tag{71}$$

For each $f \in \tilde{N}$, we check if $f$ can be approximated by $g_{\boldsymbol{u}}$ for some $\boldsymbol{u} \in \mathcal{M}_n(0, 1)$ so that $\|f - g_{\boldsymbol{u}}\|_{L^p} \leq \varepsilon$. If it can, we put one of the corresponding sequences $\boldsymbol{u}$ into $N$. By construction of $N$, we have $\log|N| \leq \log|\tilde{N}| \leq \frac{C_p}{2\varepsilon}$.

Next, we confirm $N$ is indeed a $\delta$-net of $\mathcal{M}_n(0, 1)$. Take any $\boldsymbol{u} \in \mathcal{M}_n(0, 1)$. Then, since $\tilde{N}$ is a $\varepsilon$-net of $\mathcal{M}$ and $g_{\boldsymbol{u}}$ belongs to $\mathcal{M}$, there exists $f \in \tilde{N}$ approximating $g_{\boldsymbol{u}}$ so that

$$\|g_{\boldsymbol{u}} - f\|_{L^p} \leq \varepsilon. \tag{72}$$

Now, observe that (72) implies "$f \in \tilde{N}$ can be approximated by $g_{\boldsymbol{u}}$ for some $\boldsymbol{u} \in \mathcal{M}_n(0, 1)$," so there is $\boldsymbol{v} \in N$ such that $\|f - g_{\boldsymbol{v}}\|_{L^p} \leq \varepsilon$ by the construction of $N$. So the triangle inequality implies

$$\|g_{\boldsymbol{u}} - g_{\boldsymbol{v}}\|_{L^p} \leq \|g_{\boldsymbol{u}} - f\|_{L^p} + \|f - g_{\boldsymbol{v}}\|_{L^p} \leq 2\varepsilon. \tag{73}$$

On the other hand, the left-hand side can be explicitly calculated as follows.

$$\begin{aligned}
\|g_{\boldsymbol{u}} - g_{\boldsymbol{v}}\|_{L^p} &= \left(\int_0^1 (g_{\boldsymbol{u}} - g_{\boldsymbol{v}})^p\right)^{1/p} \\
&= \left(\int_0^1 \sum_{i=1}^n (u_i - v_i)^p \mathbb{1}\left[x \in \left[\frac{i-1}{n}, \frac{i}{n}\right)\right]\right)^{1/p} \\
&= \left(\frac{1}{n} \sum_{i=1}^n (u_i - v_i i)^p\right)^{1/p} \\
&= \frac{\|\boldsymbol{u} - \boldsymbol{v}\|_p}{n^{1/p}}.
\end{aligned} \tag{74}$$

Therefore, it follows that, for any $\boldsymbol{u} \in \mathcal{M}_n(0, 1)$, there exists $\boldsymbol{v} \in N$ such that

$$\|\boldsymbol{u} - \boldsymbol{v}\|_p \leq 2n^{1/p}\varepsilon = \delta, \tag{75}$$

which proves $N$ is a $\delta$-net of $\mathcal{M}_n(0, 1)$ with respect to $p$-norm. Thus, we have

$$\log \mathcal{N}(\mathcal{M}_n(0, 1), \|\cdot\|_p, \delta) \leq \log|N| \leq \frac{C_p}{2\varepsilon} = \frac{C_p n^{1/p}}{\delta}. \tag{76}$$

$\square$

### C.1.4 LEMMA FOR PROVING SHARP ORACLE INEQUALITIES

Here, we present a general lemma that we can use to prove a sharp oracle inequality for the LSE

$$\widehat{\boldsymbol{\mu}} = \arg\min_{\boldsymbol{u}\in K}\|\boldsymbol{y} - \boldsymbol{u}\|_2^2 = \arg\min_{\boldsymbol{u}\in K}\|(\boldsymbol{\mu} + \boldsymbol{\xi}) - \boldsymbol{u}\|_2^2 \tag{77}$$

under a convex constraint $K \subset \mathbb{R}^n$ and a general noise $\boldsymbol{\xi}$. Here "sharp" means that the resulting oracle inequality has a leading constant 1. Lemma 5 below is a slight extension of the elegant argument given by Bellec (2018, Theorem 2.3), which was given for the i.i.d. Gaussian noise setting. In fact, it is essentially just a deterministic statement, so there is no requirement for the stochastic structure of the noise $\boldsymbol{\xi}$. Their key idea was to make use of convexity to obtain a stronger basic inequality than usual. Here *basic inequality* refers to the elementary fact

$$\|\widehat{\boldsymbol{\mu}} - \boldsymbol{\mu}\|_2^2 \le \|\boldsymbol{u} - \boldsymbol{\mu}\|_2^2 + 2\langle\boldsymbol{\xi}, \widehat{\boldsymbol{\mu}} - \boldsymbol{u}\rangle, \quad \forall\boldsymbol{u} \in K \tag{78}$$

that holds even if $K$ is non-convex. (78) immediately follows from the optimality of $\widehat{\boldsymbol{\mu}}$, i.e.,

$$\|\boldsymbol{y} - \widehat{\boldsymbol{\mu}}\|_2^2 \le \|\boldsymbol{y} - \boldsymbol{u}\|_2^2, \quad \forall\boldsymbol{u} \in K. \tag{79}$$

Now suppose $K$ is convex. Then, the LSE (i.e., the projection of $\boldsymbol{y}$ onto $K$) satisfies the variational inequality

$$\langle\boldsymbol{u} - \widehat{\boldsymbol{\mu}}, \boldsymbol{y} - \widehat{\boldsymbol{\mu}}\rangle \le 0, \quad \forall\boldsymbol{u} \in K, \tag{80}$$

which is an elementary result of convex geometry. Importantly, it implies

$$\|\boldsymbol{y} - \widehat{\boldsymbol{\mu}}\|_2^2 \le \|\boldsymbol{y} - \boldsymbol{u}\|_2^2 - \|\widehat{\boldsymbol{\mu}} - \boldsymbol{u}\|_2^2, \quad \forall\boldsymbol{u} \in K. \tag{81}$$

(81) can be seen as a strengthened version of (79) with the additional term $-\|\widehat{\boldsymbol{\mu}} - \boldsymbol{u}\|_2^2$. Therefore it can be used to derive a stronger version of the basic inequality (78), i.e.,

$$\|\widehat{\boldsymbol{\mu}} - \boldsymbol{\mu}\|_2^2 \le \|\boldsymbol{u} - \boldsymbol{\mu}\|_2^2 + 2\langle\boldsymbol{\xi}, \widehat{\boldsymbol{\mu}} - \boldsymbol{u}\rangle - \|\widehat{\boldsymbol{\mu}} - \boldsymbol{u}\|_2^2, \quad \forall\boldsymbol{u} \in K \tag{82}$$

This is the inequality (2.3) in Bellec (2018). Following their method, we use this fact as the starting point of the proof of Lemma 5. Recall that we do not require $\boldsymbol{\mu}$ to belong to $K$. It can be any point in $\mathbb{R}^n$, i.e., we allow model misspecification.

**Lemma 5** (Localized width and projection). *Take any $p \in [2, \infty]$. Let $\boldsymbol{\xi}, \boldsymbol{\mu} \in \mathbb{R}^n$ be arbitrarily fixed vectors and $K \subset \mathbb{R}^n$ be a convex set. Suppose that a point $\boldsymbol{u} \in K$ and positive numbers $t, s$ satisfy*

$$Z(\boldsymbol{u}, t) := \sup_{\boldsymbol{v}\in K, \|\boldsymbol{v}-\boldsymbol{u}\|_p\le t}\langle\boldsymbol{\xi}, \boldsymbol{v} - \boldsymbol{u}\rangle \le \frac{t^2}{2} + ts. \tag{83}$$

*Then, the projection $\widehat{\boldsymbol{\mu}}$ of $\boldsymbol{\mu} + \boldsymbol{\xi}$ onto $K$ satisfies*

$$\|\widehat{\boldsymbol{\mu}} - \boldsymbol{\mu}\|_2^2 \le \|\boldsymbol{u} - \boldsymbol{\mu}\|_2^2 + (t + s)^2. \tag{84}$$

*Proof.* For ease of notation, let

$$K_p(\boldsymbol{u}, t) := \{\boldsymbol{v} \in K \mid \|\boldsymbol{v} - \boldsymbol{u}\|_p \le t\}. \tag{85}$$

Note that $Z(\boldsymbol{u}, t) = \sup_{\boldsymbol{v}\in K_p(\boldsymbol{u},t)}\langle\boldsymbol{\xi}, \boldsymbol{v} - \boldsymbol{u}\rangle$. We break our analysis into two cases.

(i) If $\|\widehat{\boldsymbol{\mu}} - \boldsymbol{u}\|_p \le t$, then $\widehat{\boldsymbol{\mu}} \in K_p(\boldsymbol{u}, t)$. Therefore the basic inequality (82) implies

$$\|\widehat{\boldsymbol{\mu}} - \boldsymbol{\mu}\|_2^2 - \|\boldsymbol{u} - \boldsymbol{\mu}\|_2^2 \le 2\langle\boldsymbol{\xi}, \widehat{\boldsymbol{\mu}} - \boldsymbol{u}\rangle - \|\widehat{\boldsymbol{\mu}} - \boldsymbol{u}\|_2^2 \le 2Z(\boldsymbol{u}, t) - \|\widehat{\boldsymbol{\mu}} - \boldsymbol{u}\|_2^2 \le 2Z(\boldsymbol{u}, t). \tag{86}$$

Now use the assumption (83) to obtain

$$\|\widehat{\boldsymbol{\mu}} - \boldsymbol{\mu}\|_2^2 - \|\boldsymbol{u} - \boldsymbol{\mu}\|_2^2 \le t^2 + 2ts \le (t + s)^2. \tag{87}$$

(ii) Next, suppose $\|\widehat{\boldsymbol{\mu}} - \boldsymbol{u}\|_p > t$. Letting $\alpha := t/\|\widehat{\boldsymbol{\mu}} - \boldsymbol{u}\|_p$, we have $\alpha \in (0, 1)$. Now take $\boldsymbol{v} := \boldsymbol{u} + \alpha(\widehat{\boldsymbol{\mu}} - \boldsymbol{u})$. Then, the convexity of $K$ implies $\boldsymbol{v} \in K$, and clearly, we have

$\|\boldsymbol{v} - \boldsymbol{u}\|_p = t$, so $\boldsymbol{v}$ is a member of $K_p(\boldsymbol{u}, t)$. Therefore, we can plug $\widehat{\boldsymbol{\mu}} - \boldsymbol{u} = \alpha^{-1}(\boldsymbol{v} - \boldsymbol{u})$ into the basic inequality (82) to obtain

$$\|\widehat{\boldsymbol{\mu}} - \boldsymbol{\mu}\|_2^2 - \|\boldsymbol{u} - \boldsymbol{\mu}\|_2^2 \leq 2\langle \boldsymbol{\xi}, \widehat{\boldsymbol{\mu}} - \boldsymbol{u} \rangle - \|\widehat{\boldsymbol{\mu}} - \boldsymbol{u}\|_2^2 \tag{88}$$

$$= \frac{2}{\alpha} \langle \boldsymbol{\xi}, \boldsymbol{v} - \boldsymbol{u} \rangle - \frac{\|\boldsymbol{v} - \boldsymbol{u}\|_2^2}{\alpha^2} \tag{89}$$

$$\leq \frac{2}{\alpha} Z(\boldsymbol{u}, t) - \frac{t^2}{\alpha^2} \tag{90}$$

$$= 2 \frac{Z(\boldsymbol{u}, t)}{t} \frac{t}{\alpha} - \left(\frac{t}{\alpha}\right)^2 \tag{91}$$

$$\leq \left(\frac{Z(\boldsymbol{u}, t)}{t}\right)^2, \tag{92}$$

where we used $\|\boldsymbol{v} - \boldsymbol{u}\|_2^2 \geq \|\boldsymbol{v} - \boldsymbol{u}\|_p^2 = t^2$ in (90) and $2ab - b^2 \leq a^2$ in (92). Now, the the assumption (83) readily implies

$$\|\widehat{\boldsymbol{\mu}} - \boldsymbol{\mu}\|_2^2 - \|\boldsymbol{u} - \boldsymbol{\mu}\|_2^2 \leq \left(\frac{t}{2} + s\right)^2 \leq (t + s)^2 \tag{93}$$

Therefore the claim is true for both cases. □

### C.1.5 Risk bound for binary isotonic regression

In the sequel, we apply the general results stated in the previous sections to investigate the *binary isotonic regression* problem. In binary regression, we are given $n$ binary observations $y_i \in \{0, 1\}$, each of which is drawn independently from the Bernoulli distribution with mean $\mu_i \in [0, 1]$. The noise distribution can be described as

$$\mathbb{P}(\xi_i = 1 - \mu_i) = \mu_i, \quad \mathbb{P}(\xi_i = -\mu_i) = 1 - \mu_i. \tag{94}$$

Many calibration methods for probabilistic classification, including calibration by isotonic regression (Zadrozny and Elkan, 2002), can be seen as an instance of binary regression problems. Some authors refer to this setup as the Bernoulli model (Yang and Barber, 2019).

To the best of our knowledge, there is no previous work that investigated risk bounds in binary isotonic regression. Here, we derive a new risk bound for this setting. Recall the definitions of the isotonic cone $\mathcal{M}_n$ and its truncation $\mathcal{M}_n(a, b)$ (see (55) and (65)):

$$\mathcal{M}_n = \{\boldsymbol{u} \in \mathbb{R}^n \mid u_1 \leq \ldots \leq u_n\}, \tag{95}$$

$$\mathcal{M}_n(a, b) = \{\boldsymbol{u} \in \mathbb{R}^n \mid a \leq u_1 \leq \ldots \leq u_n \leq b\} \quad (-\infty < a < b < \infty). \tag{96}$$

From the min-max formula (57), one can observe that the unbounded set $\mathcal{M}_n$ can be replaced with the bounded closed convex set $\mathcal{M}_n(0, 1)$ in binary isotonic regression. In other words, the least squares estimator for the binary isotonic regression problem can be written as

$$\widehat{\boldsymbol{\mu}} = \underset{\boldsymbol{u} \in \mathcal{M}_n(0,1)}{\arg\min} \|\boldsymbol{y} - \boldsymbol{u}\|_2^2. \tag{97}$$

It leads to the following result.

**Proposition 2.** *With probability at least $1 - e^{-x}$, we have*

$$\frac{1}{n}\|\widehat{\boldsymbol{\mu}} - \boldsymbol{\mu}\|_2^2 \leq \min_{\boldsymbol{u} \in \mathcal{M}_n(0,1)} \frac{1}{n}\|\boldsymbol{u} - \boldsymbol{\mu}\|_2^2 + \left(\left(\frac{C}{n}\right)^{1/3} + \sqrt{\frac{2x}{n}}\right)^2, \tag{98}$$

*where $C$ is an absolute constant.*

*Remark.*

   (i) Thanks to the replacement of the unbounded set $\mathcal{M}_n$ with the bounded set $\mathcal{M}_n(0, 1)$, we do not have to go through the additional "pealing" step used in Chatterjee (2014) and Chatterjee and Lafferty (2019).

(ii) This result is valid even under model misspecification.

*Proof.* For any $\boldsymbol{u} \in \mathcal{M}_n(0,1)$ and $t > 0$, let

$$K(\boldsymbol{u}, t) := \{\boldsymbol{v} \in \mathcal{M}_n(0,1) \mid \|\boldsymbol{v} - \boldsymbol{u}\|_2 \le t\}, \tag{99}$$

$$Z(\boldsymbol{u}, t) := \sup_{\boldsymbol{v} \in K(\boldsymbol{u}, t)} \langle \boldsymbol{\xi}, \boldsymbol{v} - \boldsymbol{u} \rangle. \tag{100}$$

We first control the expectation $\mathbb{E}\left[Z(\boldsymbol{u}, t)\right]$. To this end, observe that the process $(X_{\boldsymbol{v}})_{\boldsymbol{v} \in K(\boldsymbol{u}, t)}$, where $X_{\boldsymbol{v}} := \langle \boldsymbol{\xi}, \boldsymbol{v} - \boldsymbol{u} \rangle$, is a sub-Gaussian process, i.e., for any $\boldsymbol{v}_1, \boldsymbol{v}_2 \in T$, we have $\mathbb{E}\left[X_{\boldsymbol{v}_1}\right] = 0$ and

$$\log \mathbb{E}\left[e^{\lambda(X_{\boldsymbol{v}_2} - X_{\boldsymbol{v}_1})}\right] \le \frac{\lambda^2 \|\boldsymbol{v}_2 - \boldsymbol{v}_1\|_2^2}{8}, \quad \forall \lambda \ge 0. \tag{101}$$

Now combining Dudley's chaining technique (see e.g. van Handel, 2016, Corollary 5.25) and the metric entropy bound in Theorem 4 gives

$$\begin{aligned}
\mathbb{E}\left[Z(\boldsymbol{u}, t)\right] &\le 6 \int_0^t \sqrt{\log \mathcal{N}(K(\boldsymbol{u}, t), \|\cdot\|_2, \varepsilon)} \, d\varepsilon \\
&\le 6 \int_0^t \sqrt{\frac{C_2 \sqrt{n}}{\varepsilon}} \, d\varepsilon \\
&= 12 C_2^{1/2} n^{1/4} t^{1/2},
\end{aligned} \tag{102}$$

where $C_2$ is the constant appearing in (68).

Moreover, it is straightforward to see that, for each fixed $\boldsymbol{u}$ and $t$, $Z(\boldsymbol{u}, t)$ is a convex $t$-Lipschitz function of $\boldsymbol{\xi}$. Therefore, by using Theorem 6.10 in Boucheron et al. (2013) together with (102), with probability greater than $1 - e^{-x}$, we have

$$Z(\boldsymbol{u}, t) \le \mathbb{E}\left[Z(\boldsymbol{u}, t)\right] + t\sqrt{2x} \le 12 C_2^{1/2} n^{1/4} t^{1/2} + t\sqrt{2x}. \tag{103}$$

Now, define $t^* := 4(9C_2)^{1/3} n^{1/6}$ and observe that we have $12 C_2^{1/2} n^{1/4} t^{1/2} \le \frac{t^2}{2}$ for any $t \ge t^*$. Therefore, Lemma 5 yields

$$\|\widehat{\boldsymbol{\mu}} - \boldsymbol{\mu}\|_2^2 \le \|\boldsymbol{u} - \boldsymbol{\mu}\|_2^2 + \left(t^* + \sqrt{2x}\right)^2 \le \left(\|\boldsymbol{u} - \boldsymbol{\mu}\|_2 + t^* + \sqrt{2x}\right)^2 \tag{104}$$

with probability at least $1 - e^{-x}$. we obtain the result by dividing both sides by $n$ and taking the minimum over all $\boldsymbol{u} \in \mathcal{M}_n(0,1)$. $\qquad \square$

### C.2 PROOF OF THEOREM 2

We are just one lemma away from proving our main theorem. The following lemma states that the error between the two estimates with different sets of soft labels can be upper bounded by the root-mean-square error between them.

**Lemma 6.** *For any set of soft labels $\{\eta_i'\}_{i=1}^n \in [0,1]^n$, it holds that*

$$\left|\widehat{\mathrm{Err}}^*(\eta_{1:n}') - \widehat{\mathrm{Err}}^*(\eta_{1:n})\right| \le \frac{1}{n} \sum_{i=1}^n |\eta_i' - \eta_i| \le \sqrt{\frac{1}{n} \sum_{i=1}^n (\eta_i' - \eta_i)^2}. \tag{105}$$

*Proof.* Since $x \in [0,1] \mapsto \min\{x, 1-x\}$ is 1-Lipschitz, we have

$$\left|\widehat{\mathrm{Err}}^*(\eta_{1:n}') - \widehat{\mathrm{Err}}^*(\eta_{1:n})\right| \le \frac{1}{n} \sum_{i=1}^n |\min\{\eta_i', 1 - \eta_i'\} - \min\{\eta_i, 1 - \eta_i\}| \tag{106}$$

$$\le \frac{1}{n} \sum_{i=1}^n |\eta_i' - \eta_i|. \tag{107}$$

The second inequality follows from Jensen's inequality.

$\qquad \square$

We can finally prove Theorem 2.

*Proof.* By the triangle inequality, we have

$$\left|\widehat{\mathrm{Err}^*}(\widehat{\eta}_{1:n}^{\mathrm{iso}}) - \mathrm{Err}^*\right| \leq \left|\widehat{\mathrm{Err}^*}(\widehat{\eta}_{1:n}^{\mathrm{iso}}) - \widehat{\mathrm{Err}^*}(\eta_{1:n})\right| + \left|\widehat{\mathrm{Err}^*}(\eta_{1:n}) - \mathrm{Err}^*\right|. \tag{108}$$

Using Lemma 6 for the first term and Proposition 3.2 of Ishida et al. (2023) for the second term, we have

$$\left|\widehat{\mathrm{Err}^*}(\widehat{\eta}_{1:n}^{\mathrm{iso}}) - \mathrm{Err}^*\right| \leq \sqrt{\frac{1}{n}\sum_{i=1}^{n}\left(\widehat{\eta}_i^{\mathrm{iso}} - \eta_i\right)^2} + \sqrt{\frac{\log(4/\delta)}{8n}} \tag{109}$$

with probability at least $1 - \delta/2$. Now, we evaluate the first term on the right-hand side by applying Proposition 2 for $\boldsymbol{\mu} = (\eta_{(1)}, \ldots, \eta_{(n)})$. Conditioned on $\{x_i\}_{i=1}^n$, with probability at least $1 - \delta/2$, we have

$$\frac{1}{n}\sum_{i=1}^{n}\left(\widehat{\eta}_i^{\mathrm{iso}} - \eta_i\right)^2 \leq \min_{\boldsymbol{u}\in\mathcal{M}_n(0,1)}\frac{1}{n}\sum_{i=1}^{n}\left(u_i - \eta_{(i)}\right)^2 + \left(\left(\frac{C}{n}\right)^{1/3} + \sqrt{\frac{2\log(2/\delta)}{n}}\right)^2. \tag{110}$$

Since $f$ is increasing and $\tilde{\eta}_{(1)} = f(\eta_{(1)}) \leq \cdots \leq \tilde{\eta}_{(n)} = f(\eta_{(n)})$, we have $\eta_{(1)} \leq \cdots \leq \eta_{(n)}$, i.e., $(\eta_{(1)}, \ldots, \eta_{(n)}) \in \mathcal{M}_n$. Therefore, $\min_{\boldsymbol{u}\in\mathcal{M}_n(0,1)}\frac{1}{n}\sum_{i=1}^{n}\left(u_i - \eta_{(i)}\right)^2$ is zero as we can choose $\boldsymbol{u} = (\eta_{(1)}, \ldots, \eta_{(n)})$. As a result, we have

$$\sqrt{\frac{1}{n}\sum_{i=1}^{n}\left(\widehat{\eta}_i^{\mathrm{iso}} - \eta_i\right)^2} \leq \left(\frac{C}{n}\right)^{1/3} + \sqrt{\frac{2\log(2/\delta)}{n}}. \tag{111}$$

Plugging (111) into (109) and rewriting $C^{1/3}$ as $C$, we have

$$\left|\widehat{\mathrm{Err}^*}(\widehat{\eta}_{1:n}^{\mathrm{iso}}) - \mathrm{Err}^*\right| \leq \frac{C}{n^{1/3}} + 2\sqrt{\frac{2\log(2/\delta)}{n}} \tag{112}$$

with probability at least $1 - \delta$. $\qquad\square$

We can also prove an extension to the case where random noise is added to the corrupted soft labels.

*Proof of Theorem 3.* By the same argument as in the main theorem proof, we have

$$\left|\widehat{\mathrm{Err}^*}(\widehat{\eta}_{1:n}^{\mathrm{iso}}) - \mathrm{Err}^*\right| \leq \sqrt{\frac{1}{n}\sum_{i=1}^{n}\left(\widehat{\eta}_i^{\mathrm{iso}} - \eta_i\right)^2} + \sqrt{\frac{\log(6/\delta)}{8n}} \tag{113}$$

with probability at least $1 - \delta/3$. Also, conditioned on $\{x_i\}_{i=1}^n$, with probability at least $1 - \delta/3$, we have

$$\frac{1}{n}\sum_{i=1}^{n}\left(\widehat{\eta}_i^{\mathrm{iso}} - \eta_i\right)^2 \leq \min_{\boldsymbol{u}\in\mathcal{M}_n(0,1)}\frac{1}{n}\sum_{i=1}^{n}\left(u_i - \eta_{(i)}\right)^2 + \left(\left(\frac{C}{n}\right)^{1/3} + \sqrt{\frac{2\log(3/\delta)}{n}}\right)^2. \tag{114}$$

Now, under the new assumption, $f$ has an inverse function $f^{-1}$, which is also differentiable and increasing. Therefore, the mean value theorem gives

$$\eta_{(i)} = f^{-1}(\tilde{\eta}_{(i)}) - \frac{\varepsilon_{(i)}}{f'(t_{(i)})} \tag{115}$$

for some $t_{(i)} \in (0,1)$, which means

$$\min_{\boldsymbol{u}\in\mathcal{M}_n(0,1)}\frac{1}{n}\sum_{i=1}^{n}(u_i - \eta_{(i)})^2 \leq \frac{2}{n}\min_{\boldsymbol{u}\in\mathcal{M}_n(0,1)}\sum_{i=1}^{n}(u_i - f^{-1}(\tilde{\eta}_{(i)}))^2 + \frac{2}{n}\sum_{i=1}^{n}\frac{\varepsilon_{(i)}^2}{f'(t_{(i)})^2}. \tag{116}$$

Since $f^{-1}$ is increasing and $\eta_{\widetilde{(1)}} \leq \cdots \leq \eta_{\widetilde{(n)}}$, the first term vanishes by choosing $u_i = f^{-1}(\tilde{\eta}_{(i)})$. By using the assumption that $f' \geq c$, we obtain

$$\min_{\boldsymbol{u} \in \mathcal{M}_n(0,1)} \frac{1}{n} \sum_{i=1}^n (u_i - \eta_{(i)})^2 \leq \frac{2}{c^2 n} \sum_{i=1}^n \varepsilon_{(i)}^2. \tag{117}$$

Since $\varepsilon_{(i)}^2 \in [0, 1]$ and $\mathbb{E}\left[\varepsilon_{(i)}^2\right] \leq \sigma^2$ for each $i$, the Hoeffding's inequality gives

$$\frac{1}{n} \sum_{i=1}^n \varepsilon_{(i)}^2 \leq \sigma^2 + \sqrt{\frac{\log(3/\delta)}{2n}} \tag{118}$$

with probability at least $1 - \delta/3$.

By combining the above bounds, there exists a constant $C' > 0$ such that

$$\left| \widehat{\mathrm{Err}^*}(\hat{\eta}_{1:n}^{\mathrm{iso}}) - \mathrm{Err}^* \right| \leq C' \left( \sigma + \frac{1}{n^{1/3}} + \sqrt{\frac{\log(1/\delta)}{n}} \right) \tag{119}$$

holds with probability at least $1 - \delta$. $\qquad\square$

## D    EXPERIMENTAL DETAILS

We utilized the scikit-learn library (Pedregosa et al., 2011), version 1.6.1, for isotonic regression. We used the implementation of the histogram binning algorithm provided by the `uncertainty-calibration` package (version 0.1.4; Kumar et al., 2019). We employed the beta-calibration implementation provided in the `betacal` package (version 1.1.0; Kull et al., 2017). We used the `bootstrap` function from the SciPy library (Virtanen et al., 2020), version 1.15.3, to obtain 95% bootstrap confidence intervals. For each estimation method, the experiment took around 20–30 minutes on a CPU.

For the sake of comparison in Fig. 4c, we trained a ResNet-18 (He et al., 2016) on Fashion-MNIST for 100 epochs with a batch size of 128 using the Adam optimizer with a learning rate of 0.001. It took less than an hour.

Our experiments do not require any special computer resouces. All of them were conducted on the CPU of a single Apple MacBook Pro (M1 chip, 16GB RAM) except for the ResNet training where we used a T4 GPU on Google Colab.

### D.1    CORRUPTION PARAMETERS

In experiments with synthetic mixture-of-gaussians data, we used the following corruption function:

$$f(p; a, b) = \left( 1 + \left( \frac{1-p}{p} \right)^{1/a} \frac{1-b}{b} \right)^{-1}, \quad 0 < p < 1, \ a > 0, \ 0 < b < 1. \tag{120}$$

Fig. 8 shows the graph of $f(p; a, b)$ for various values of the parameters $a$ and $b$. As you can see, the parameter $a$ makes the soft labels over-confident when $a < 1$, leading to an underestimation of the Bayes error, and under-confident when $a > 1$, causing an overestimation. On the other hand, setting $b$ to values other than $0.5$ results in asymmetric, skewed corruption.

We conducted the same experiment as in Fig. 4a for various values of $a$ and $b$. The results are shown in Fig. 9. Our calibration-based estimators consistently succeed in preventing over- or underestimation of the Bayes error across all sets of parameter values.

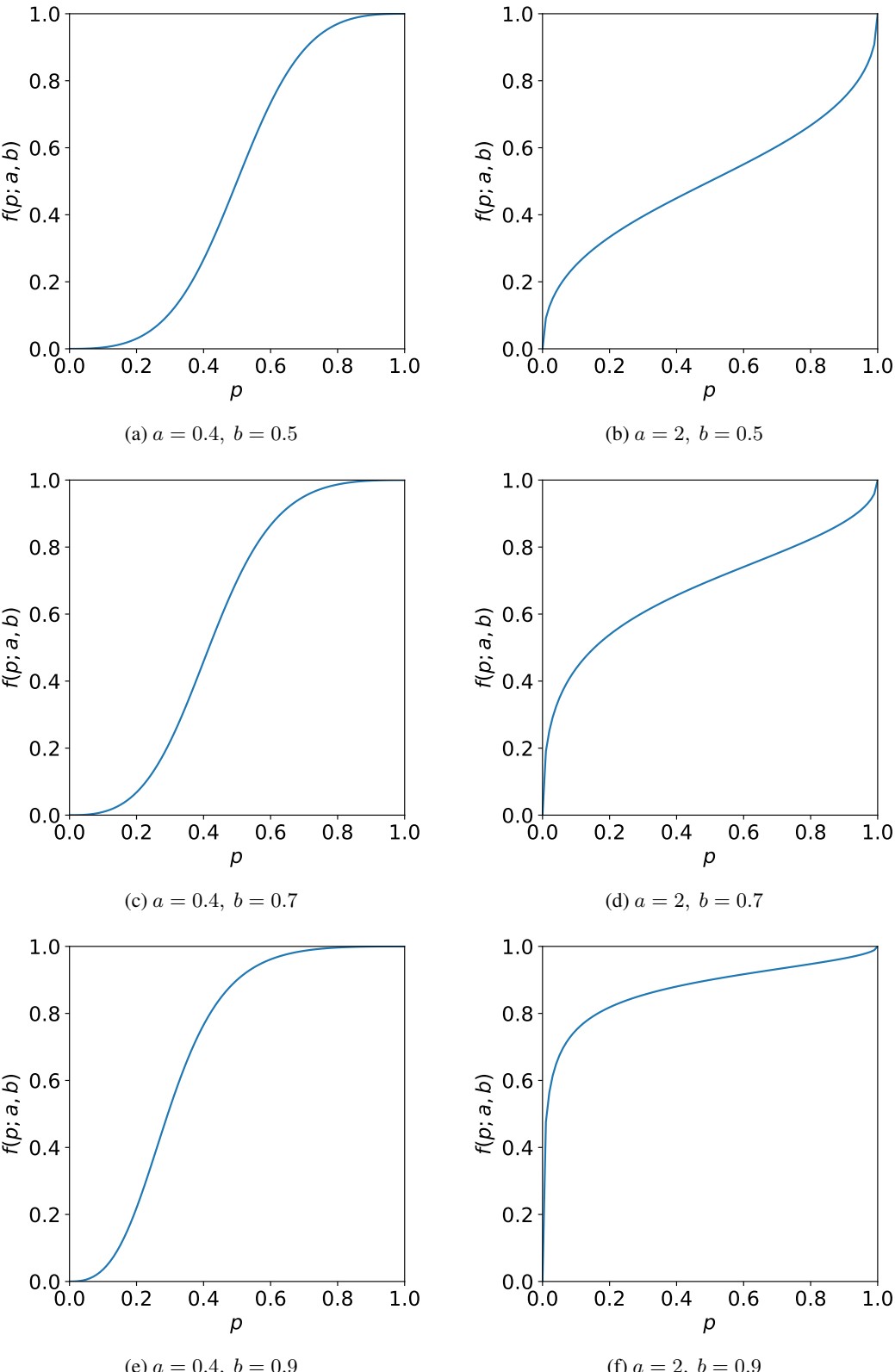

Figure 8: The corruption function $f(p; a, b)$ for various parameters $a$ and $b$.

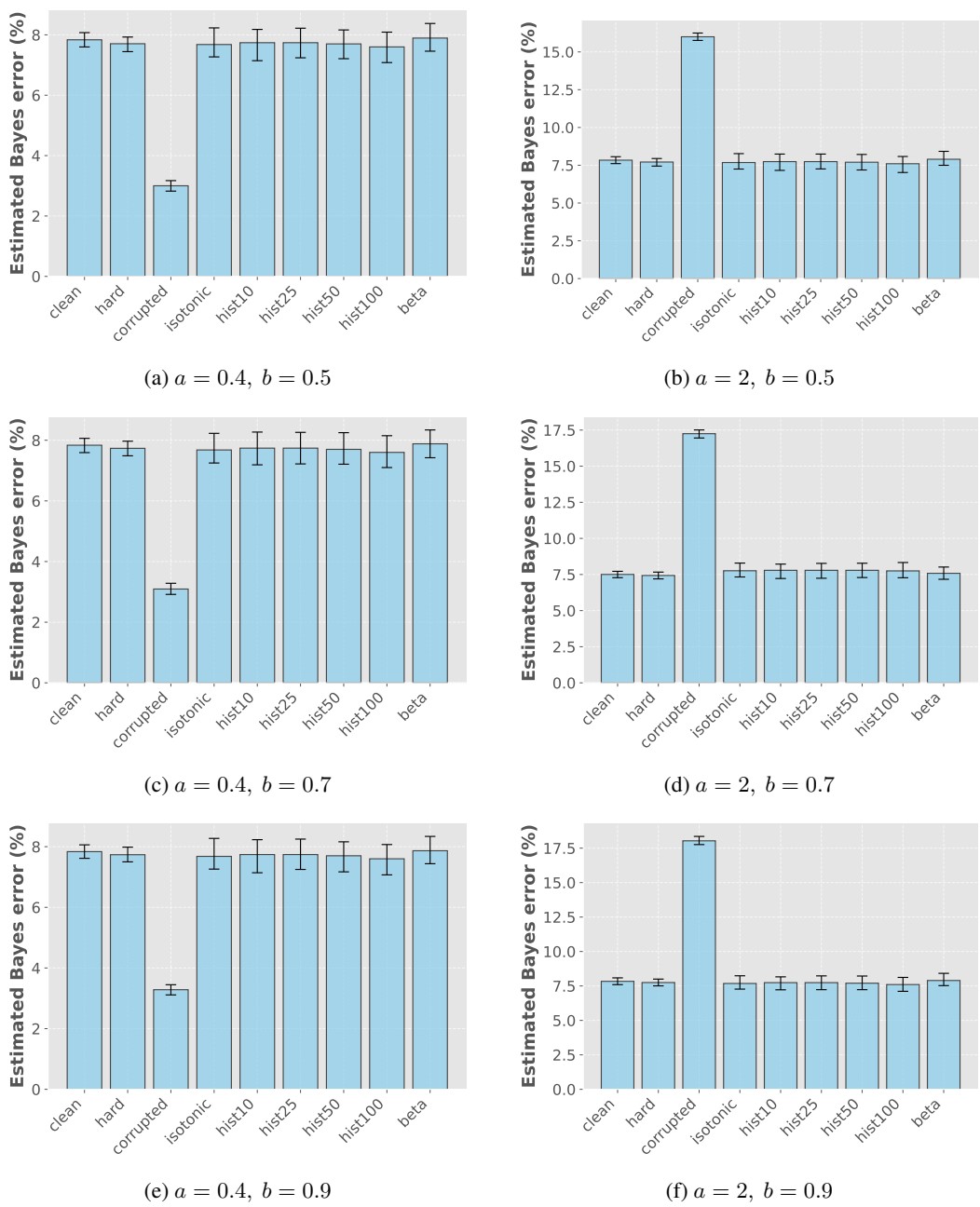

Figure 9: The estimated Bayes error for the synthetic dataset with various corruption parameters $a$ and $b$.

## D.2 Violation of the assumption of Theorem 3

In Section 3, we presented Theorem 3, which provides a theoretical guarantee for our Bayes error estimator based on isotonic calibration when the corruption is noisy. This result assumes that the derivative of the function $f$ satisfies $f' \geq c$ for some *strictly positive* constant $c$. Theoretically, it is still unclear what happens when this assumption is violated, i.e., when $f'$ can be arbitrarily close to zero. Here we empirically investigate the effects of such a violation using synthetic data.

### D.2.1 Experimental settings

We draw $n = 10000$ data points $\{x_i\}_{i=1}^n$ from a Gaussian mixture $\mathbb{P}_{\mathcal{X}} = 0.6 \cdot \mathbb{P}_0 + 0.4 \cdot \mathbb{P}_1$, where $\mathbb{P}_0 = N((0,0), I_2)$, $\mathbb{P}_1 = N((2,2), I_2)$.[11] For each data point $x_i$, we generate its soft label $\tilde{\eta}_i$ by sampling $m = 3, 5, 10, 25, 50, 100$ hard labels from the corrupted posterior distribution $\mathrm{Bern}(f(\eta(x_i); a, b))$ [12] and taking their average. In other words, we obtain the soft label for $x_i$ as a draw from $\mathrm{Binom}(m, f(\eta(x_i); a, b))$ divided by $m$. by drawing from $\mathrm{Binom}(m, f(\eta(x_i); a, b))$ and dividing the result by $m$. By changing the number $m$ of hard labels, we can create different noise levels $\sigma = O(\frac{1}{\sqrt{m}})$: the smaller $m$ gets, the greater the noise level becomes. We consider two sets of corruption parameters: $(a, b) = (2, 0.5)$ and $(a, b) = (0.4, 0.5)$. As we saw in Appendix D.1, the former corresponds to under-confident soft labels, and the latter corresponds to over-confident soft labels. Note that the former satisfies the assumption of Theorem 3 while the latter does not, i.e., the derivative $f'$ can be arbitrarily small. You can see this visually in Fig. 8. Then, we estimate the Bayes error from these corrupted soft labels $\{\tilde{\eta}_i\}_{i=1}^n$ using the following methods (which we used in Section 4.1):

(i) `corrupted`: the estimator with corrupted soft labels, i.e., $\widehat{\mathrm{Err}}^*(\tilde{\eta}_{1:n})$.

(ii) The estimator with soft labels obtained by calibrating the corrupted soft labels. We use the following calibration algorithms: isotonic calibration (`isotonic`), uniform-mass histogram binning with $10, 25, 50$ and $100$ bins (`hist10`, `hist25`, `hist50` and `hist100`), and beta calibration (`beta`).

As in Section 4, we use 1000 bootstrap resamples to compute a 95% confidence interval for each method.

### D.2.2 Results

Fig. 10 shows the estimated Bayes error for various numbers $m$ of hard labels per data point. The black dashed lines indicate the Bayes error estimated with clean soft labels, which is expected to be a good approximation of the true Bayes error. All the non-parametric calibration methods (`isotonic` and `hist*`) perform similarly. However, parametric beta calibration (`beta`) performs poorly.

Particularly, when $(a, b) = (0.4, 0.5)$ and the assumption of Theorem 3 is violated, the more we add hard labels per data point the worse the estimation performance gets even though the noise level decreases. In the large-$m$ (i.e., small noise level) regime, beta calibration goes "too far" and consistently overestimates the Bayes error. On the other hand, isotonic calibration and histogram binning perform consistently well, even for relatively small $m$. When $(a, b) = (2, 0.5)$ and the assumption of Theorem 3 is satisfied, the performance of all the methods tends to improve as the noise level decreases. However, isotonic calibration and histogram binning still outperform beta calibration, especially for small $m$.

These results suggest that isotonic calibration and histogram binning are relatively robust to corruption functions with small derivative, whereas beta calibration is not. It is interesting that beta calibration performs so poorly when it is a *well-specified* parametric model in a sense, i.e., the corruption function $f$ is an inverse function of the beta calibration map. This fact suggests that choosing appropriate calibration methods, such as isotonic calibration, is crucial in our algorithm design.

Another interesting thing is that `corrupted`'s performance becomes worse as the number $m$ of hard labels per data point increases when $(a, b) = (2, 0.5)$. This would be because, as we sample

---

[11]This is the same distribution as we used in Section 4.1.

[12]$\mathrm{Bern}(p)$ is the Bernoulli distribution with mean $p$.

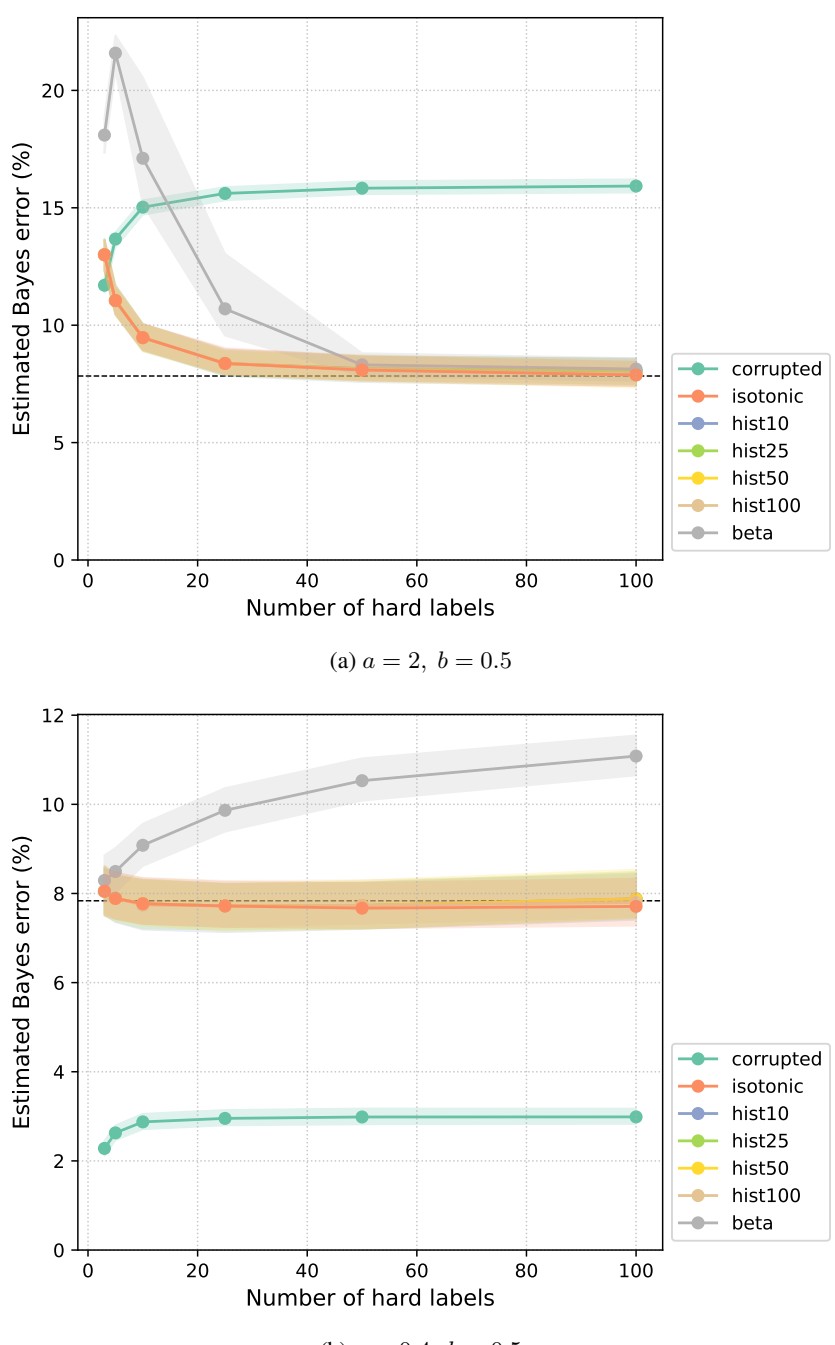

(a) $a = 2, \; b = 0.5$

(b) $a = 0.4, \; b = 0.5$

Figure 10: The Bayes error estimated by directly using corrupted soft labels (`corrupted`) and by calibrating them (others) for various numbers $m$ of hard labels per data point. The 95% bootstrap confidence intervals are shown as shaded regions around each line. The black dashed lines indicate the Bayes error estimated with clean soft labels, which is expected to be a good approximation of the true Bayes error.

more hard labels, the resulting soft labels are pulled towards the corrupted posterior mean, which makes them more biased. This result highlights the need to calibrate soft labels.

### D.3 FURTHER EXPERIMENTS

In this section, we present additional details and results of the experiments mentioned in Section 4.2.

#### D.3.1 ICLR PEER-REVIEW DATASETS

We put together new datasets, which consist of $n = 32,829$ instances of peer-review results for the past ICLR conferences. Peer-review can be considered as a binary classification task (accept/reject). We used our datasets to estimate the Bayes error of the ICLR reviews, which is the probability that the ideal, most competent possible reviewer mistakenly rejects a good paper or accepts a bad paper. It can be regarded as representing the inherent difficulty of the review task.

For each paper submission $x_i$, we utilized the OpenReview API to retrieve:

- Scores $s_i^{(j)}$ and confidences $c_i^{(j)}$ by the reviewers ($j = 1, \ldots, \#\text{reviewers}$)
- The final decision $y_i$: accept ($y_i = 1$) or reject ($y_i = 0$)

The averaged score $s_i$ is calculated as $s_i = \frac{\sum_j c_i^{(j)} s_i^{(j)}}{\sum_j c_i^{(j)}}$. We can then obtain a soft label $\tilde{\eta}_i$ for $x_i$ by normalizing the averaged score $s_i$ to fit into $[0, 1]$. Of course, this soft label $\tilde{\eta}_i$ should be considered as corrupted, so we apply isotonic calibration (and other calibration algorithms) before using them to estimate the Bayes error.

We merged data from ICLR 2017–2025 to construct a dataset consisting of $n = 32,829$ examples, each of which has a corrupted soft label (i.e., normalized average score) and a single hard label (i.e., final decision) for calibration. We also conducted experiments with single-year datasets (ICLR2017, ..., ICLR2025).

#### D.3.2 VARIANTS OF BETA CALIBRATION

Beta calibration is a calibration method with three adjustable parameters $a, b, m$, and this is what we have been using as `beta` since the initial submission. The `beta-*` variants are beta calibration with restricted parameters, which were also mentioned in the original beta calibration paper (Kull et al., 2017):

- `beta-am`: $a$ and $m$ are adjustable; $b$ is fixed to $b = a$.
- `beta-ab`: $a$ and $b$ are adjustable; $m$ is fixed to $m = 1/2$.
- `beta-a`: only $a$ is adjustable; $b, m$ is fixed to $b = a, m = 1/2$.

#### D.3.3 RESULTS

The full results are shown in Fig. 11p. For ChaosNLI datasets (SNLI, MNLI, AbductiveNLI), their GitHub repository provides predictions of some pre-trained models. As a reference value, we show the best error rates among them as the dashed horizontal lines in Figures 11d to 11f. A problem in these experiments is that, especially for real-world (non-synthetic) datasets, it is hard to decide which calibration algorithm is the best, since the true Bayes error is unknown. We will solve this problem in Appendix D.4.

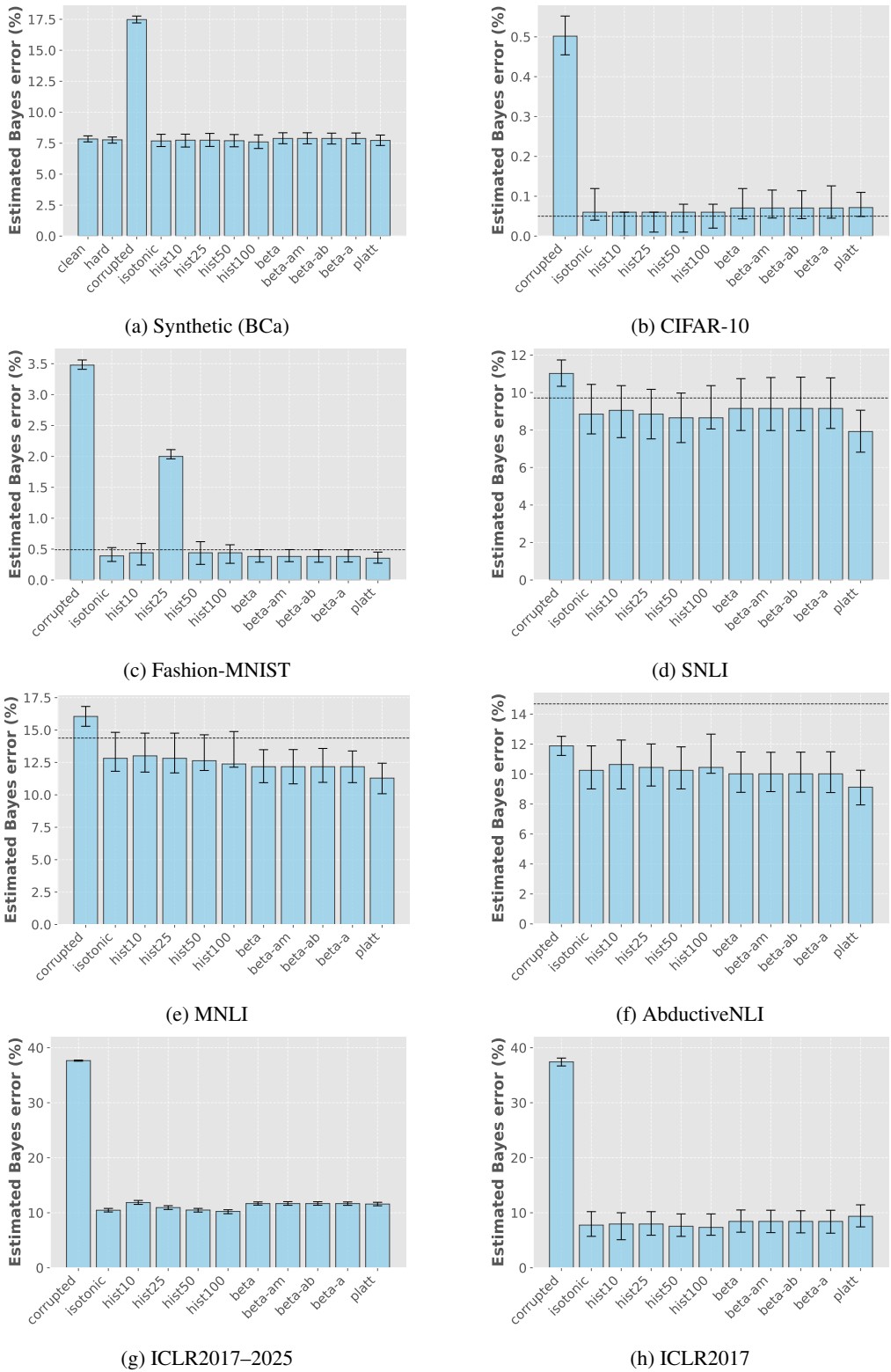

Figure 11: Estimated Bayes error across various calibration algorithms and datasets.

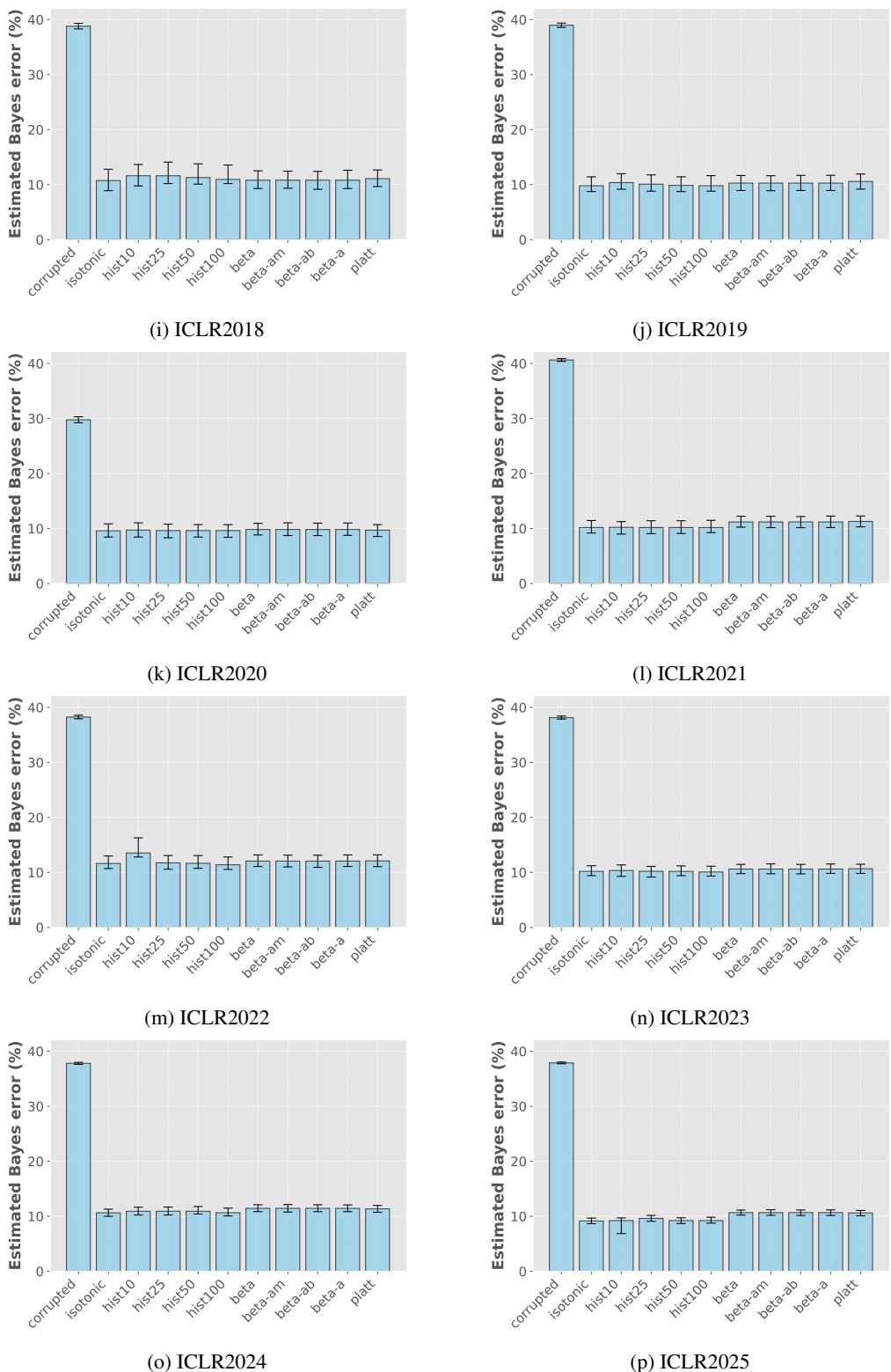

Figure 11: Estimated Bayes error across various calibration algorithms and datasets (continued).

### D.4 EVALUATING CALIBRATION ALGORITHMS AGAINST REAL-WORLD DATASETS WITH FEEBEE

In this section, we present the full details and results of the FeeBee experiment described in Section 4.3. We first review FeeBee (Renggli et al., 2021), a real-world evaluation framework for Bayes error estimators. Then, we present experimental results of various calibration algorithms evaluated using FeeBee.

#### D.4.1 REVIEW OF FEEBEE

In the long history of the field of Bayes error estimation, evaluation of estimators on real-world datasets has been a challenging task. For synthetic datasets, one can easily compute exact or approximate Bayes error rates; however, for real-world datasets, it is practically impossible to obtain ground-truth Bayes error rates as they depend on the unknown data distribution. Of course, it is trivial to obtain a lower bound and an upper bound of the Bayes error rate: if we have a classifier with error rate $E$ on a dataset, then the Bayes error rate should be somewhere between $0$ and $E$. However, such bounds are not informative enough. For example, constant estimators that always return any value between $0$ and $E$ are technically valid from this perspective, but they are obviously useless in practice.

To address this issue, Renggli et al. (2021) proposed an evaluation framework called FeeBee. The key idea of FeeBee is to generate a series of datasets from a given real-world dataset by injecting various levels of synthetic label noise. To be more specific, for a noise level $\rho \in [0, 1]$, FeeBee generates a new dataset by replacing each original label $Y$ with a uniformly random label $U$ with probability $\rho$:

$$Y_\rho := Z \cdot U + (1 - Z) \cdot Y, \tag{121}$$

where $Z \sim \text{Bernoulli}(\rho)$. Importantly, there is a simple relationship between the Bayes error rates $\text{Err}^*$ on the original dataset and $\text{Err}^*_\rho$ on the noise-injected dataset:

$$\text{Err}^*_\rho = \rho \cdot \frac{1}{2} + (1 - \rho) \cdot \text{Err}^*. \tag{122}$$

Since $0 \leq \text{Err}^* \leq E$, we can derive the following bounds on $\text{Err}^*_\rho$:

$$L(\rho) := \frac{\rho}{2} \leq \text{Err}^*_\rho \leq \frac{\rho}{2} + (1 - \rho)E =: U(\rho). \tag{123}$$

Based on this observation, FeeBee first generates many noise-injected datasets with different noise levels $\rho \in [0, 1]$, and then evaluates a given Bayes error estimator on each of them. Ideally, the resulting estimates $\widehat{\text{Err}^*_\rho}$ should lie within the bounds $[L(\rho), U(\rho)]$ given in (123). If the estimates fall outside the bounds, we penalize the estimator by the amount of violation. By aggregating the penalties over all noise levels, we can obtain a single score for the estimator on the given real-world dataset:

$$\text{FeeBee} := \int_0^1 \left[ \left( \widehat{\text{Err}^*_\rho} - U(\rho) \right)_+ + \left( L(\rho) - \widehat{\text{Err}^*_\rho} \right)_+ \right] d\rho, \tag{124}$$

where $(x)_+ := \max\{x, 0\}$. In practice, the integral can be approximated by a finite sum: for a sufficiently large $N \in \mathbb{N}$, the approximate FeeBee score can be computed as

$$\text{FeeBee} \approx \frac{1}{N+1} \sum_{i=0}^N \left[ \left( \widehat{\text{Err}^*_{\rho_i}} - U(\rho_i) \right)_+ + \left( L(\rho_i) - \widehat{\text{Err}^*_{\rho_i}} \right)_+ \right], \tag{125}$$

where $\rho_i := \frac{i}{N}$ ($i = 0, 1, \ldots, N$). The lower the FeeBee score is, the better the estimator is. FeeBee provides a practical way to evaluate Bayes error estimators on real-world datasets without requiring knowledge of the true Bayes error rates.

#### D.4.2 COMPARING CALIBRATION ALGORITHMS USING FEEBEE

Here, we present experimental results where various calibration algorithms are evaluated using the FeeBee framework. We use the following real-world datasets: CIFAR-10/CIFAR-10H, Fashion-MNIST/Fashion-MNIST-H, SNLI, MNLI, AbductiveNLI, ICLR2017–2025, ICLR2017, . . . , and

Table 2: FeeBee scores of calibration algorithms across real-world datasets (lower is better). The best scores for each dataset are highlighted in **bold**, and the rank within each column is shown in parentheses.

| | Dataset | | | | |
|---|---|---|---|---|---|
| Algorithm | CIFAR-10 | Fashion-MNIST | SNLI | MNLI | AbductiveNLI |
| isotonic | 0.00307 (3) | **0.00240 (1)** | 0.00292 (2) | 0.00147 (2) | 0.00118 (2) |
| hist-10 | 0.00318 (4) | 0.00250 (2) | **0.00283 (1)** | 0.00210 (5) | 0.00143 (3) |
| hist-25 | **0.00253 (1)** | 0.00825 (6) | 0.00356 (4) | 0.00322 (7) | 0.00362 (4) |
| hist-50 | 0.00322 (5) | 0.00329 (4) | 0.00520 (5) | 0.00421 (9) | 0.00558 (5) |
| hist-100 | 0.00329 (6) | 0.00373 (5) | 0.00714 (6) | 0.00666 (10) | 0.00813 (6) |
| beta | 0.02396 (8) | 0.08796 (8) | 0.01392 (8) | 0.00380 (8) | 0.05400 (9) |
| beta-am | 0.02401 (10) | 0.09055 (10) | 0.01452 (9) | 0.00208 (4) | 0.05204 (8) |
| beta-ab | 0.02335 (7) | 0.08737 (7) | 0.01476 (10) | **0.00143 (1)** | 0.04949 (7) |
| beta-a | 0.02400 (9) | 0.08878 (9) | 0.01370 (7) | 0.00223 (6) | 0.05537 (10) |
| platt | 0.00278 (2) | 0.00262 (3) | 0.00305 (3) | 0.00154 (3) | **0.00081 (1)** |

| | Dataset | | | | |
|---|---|---|---|---|---|
| Algorithm | ICLR2017-2025 | ICLR2017 | ICLR2018 | ICLR2019 | ICLR2020 |
| isotonic | 0.00013 (5) | 0.00508 (7) | 0.00166 (6) | 0.00120 (6) | 0.00085 (5) |
| hist-10 | 0.00011 (3) | 0.00469 (6) | 0.00232 (7) | 0.00160 (7) | 0.00106 (6) |
| hist-25 | 0.00016 (7) | 0.01086 (8) | 0.00517 (8) | 0.00352 (8) | 0.00137 (8) |
| hist-50 | 0.00033 (9) | 0.01829 (9) | 0.00863 (9) | 0.00720 (9) | 0.00162 (9) |
| hist-100 | 0.00056 (10) | 0.02915 (10) | 0.01586 (10) | 0.01114 (10) | 0.00181 (10) |
| beta | 0.00012 (4) | 0.00109 (2) | 0.00060 (2) | 0.00039 (2) | 0.00108 (7) |
| beta-am | 0.00014 (6) | 0.00120 (3) | 0.00062 (3) | **0.00035 (1)** | 0.00078 (4) |
| beta-ab | 0.00010 (2) | **0.00082 (1)** | **0.00049 (1)** | 0.00108 (5) | 0.00076 (3) |
| beta-a | 0.00017 (8) | 0.00200 (5) | 0.00097 (5) | 0.00066 (4) | 0.00056 (2) |
| platt | **0.00001 (1)** | 0.00135 (4) | 0.00067 (4) | 0.00059 (3) | **0.00030 (1)** |

| | Dataset | | | | |
|---|---|---|---|---|---|
| Algorithm | ICLR2021 | ICLR2022 | ICLR2023 | ICLR2024 | ICLR2025 |
| isotonic | 0.00098 (6) | 0.00064 (6) | 0.00034 (6) | 0.00022 (6) | 0.00024 (6) |
| hist-10 | 0.00156 (7) | 0.00090 (7) | 0.00074 (7) | 0.00044 (7) | 0.00042 (7) |
| hist-25 | 0.00183 (8) | 0.00205 (8) | 0.00143 (8) | 0.00090 (8) | 0.00060 (8) |
| hist-50 | 0.00388 (9) | 0.00386 (9) | 0.00265 (9) | 0.00133 (9) | 0.00093 (9) |
| hist-100 | 0.00621 (10) | 0.00684 (10) | 0.00509 (10) | 0.00270 (10) | 0.00158 (10) |
| beta | 0.00042 (4) | 0.00023 (3) | 0.00030 (5) | 0.00005 (2) | 0.00009 (3) |
| beta-am | 0.00026 (3) | 0.00031 (5) | 0.00020 (2) | 0.00006 (3) | 0.00015 (5) |
| beta-ab | 0.00019 (2) | 0.00021 (2) | 0.00021 (3) | **0.00003 (1)** | 0.00007 (2) |
| beta-a | **0.00017 (1)** | **0.00010 (1)** | **0.00008 (1)** | 0.00006 (4) | **0.00007 (1)** |
| platt | 0.00046 (5) | 0.00029 (4) | 0.00023 (4) | 0.00011 (5) | 0.00011 (4) |

ICLR2025. For each dataset, we compare the FeeBee scores of the following calibration algorithms: isotonic calibration (isotonic), histogram binning (hist-10, hist-25, hist-50 and hist-100), full three-parameter beta calibration (beta), beta calibration with $b = a$ (beta-am), beta calibration with $m = \frac{1}{2}$ (beta-ab), beta calibration with $b = a, m = \frac{1}{2}$ (beta-a), and Platt scaling (platt). We set $N = 100$ for the approximation of FeeBee scores.

**Choosing $E$**    To compute the FeeBee scores, we need to choose a classifier error rate $E$ for each dataset. For image classification datasets (CIFAR-10 & Fashion-MNIST), we use the error rates of Vision Transformer (ViT) models as we have seen in Section 4. For natural language inference datasets (SNLI, MNLI & AbductiveNLI), the ChaosNLI GitHub repository provides predictions of some pre-trained models. We use the best error rates among them as $E$. For the ICLR peer-review datasets, we do not have any pre-trained models, so we simply set $E$ to the overall acceptance rate, which is the error rate of a trivial reviewer who rejects any given paper no matter what.

**Results**    The full results are shown in Table 2.

### D.5 ORDER BREAKAGE

Theorem 2 and Theorem 3 assume that the corruption function $f$ is order-preserving although the latter theorem allows random noise to be added after the order-preserving transformation. To analyze how much the estimation performance degrades when the order-preserving assumption is violated, we conducted experiments on synthetic datasets where we can control the degree of order breakage.

Let $f$ be the corruption function used in Section 4. We define a new, non-order-preserving corruption $f_\sigma$ as follows:

$$f_\sigma(\eta) = \text{sigmoid}(\text{logit}(f(\eta)) + z), \quad \text{where } z \sim \mathcal{N}(0, \sigma^2). \tag{126}$$

Here, $\sigma$ controls the amount of fluctuation added after the order-preserving transformation $f$. By increasing $\sigma$, we can increase the degree of order breakage. We first consider the case where corrupted soft labels are obtained as $\tilde{\eta}_i = f_\sigma(\eta_i)$. We also consider the "non-monotonic skew + random noise" setting, i.e., we obtain corrupted soft labels as an average of $m$ independent hard labels sampled from posteriors skewed by the above non-order-preserving corruption:

$$\tilde{\eta}_i = \frac{1}{m} \sum_{j=1}^{m} y_i^{(j)}, \tag{127}$$

$$\text{where } y_i^{(j)} \sim \text{Bernoulli}\left(f_\sigma(\eta_i)\right), \quad z \sim \mathcal{N}(0, \sigma^2). \tag{128}$$

To quantify the degree of order breakage, we use the Kendall tau (Kendall, 1938) between $\eta_i$ and $f_\sigma(\eta_i)$. The Kendall tau or Kendall's rank correlation coefficient is a non-parametric measure of ordinal correspondence or monotonicity between two variables. It takes values in $[-1, 1]$. If the relationship between two variables is completely increasing, the Kendall tau becomes 1. If they are in a completely decreasing relationship, it takes a value of $-1$. Given the Kendall tau $\tau$, the probability of order breakage (in our case, the frequency that we have $\eta_i \leq \eta_j$, $f_\sigma(\eta_i) > f_\sigma(\eta_j)$ or vice versa for a randomly picked pair $i < j$) can be obtained as $\frac{1-\tau}{2}$.

We conducted experiments as below using the same Gaussian mixture model as in Section 4. For various order breakage levels $\sigma = 10^{-10}, 10^{-9}, \ldots, 10^0$, we estimated the Bayes error from a dataset containing $n = 10,000$ corrupted soft labels generated as $\tilde{\eta}_i = f_\sigma(\eta_i)$ or from $m = 50$ hard labels sampled from posteriors skewed by $f_\sigma$. Fig. 12 shows the estimated Bayes error as a function of the Kendall tau between $\eta_i$ and $f_\sigma(\eta_i)$. The black dashed line indicates the Bayes error estimated using the clean/true soft labels and is supposed to be a good approximation of the true Bayes error.

As expected, the estimation performance degrades as the degree of order breakage increases (i.e., as the Kendall tau decreases). However, in the noiseless setting (Fig. 12a), all the estimators produced estimates almost indistinguishable from the true Bayes error when the Kendall tau is sufficiently large (say, when $\tau \geq 0.95$ or when the order breakage probability is less than $2.5\%$). For the noisy setting (Fig. 12b), the results are a bit different. Overall, the estimation performance improves as the Kendall tau increases, but for beta calibration and its variants, the estimates never get very close to the true Bayes error even when the Kendall tau is nearly 1. Other calibration methods, including isotonic calibration, produced estimates fairly close to the true Bayes error (but not as close as in the noiseless setting) for sufficiently high Kendall tau values.

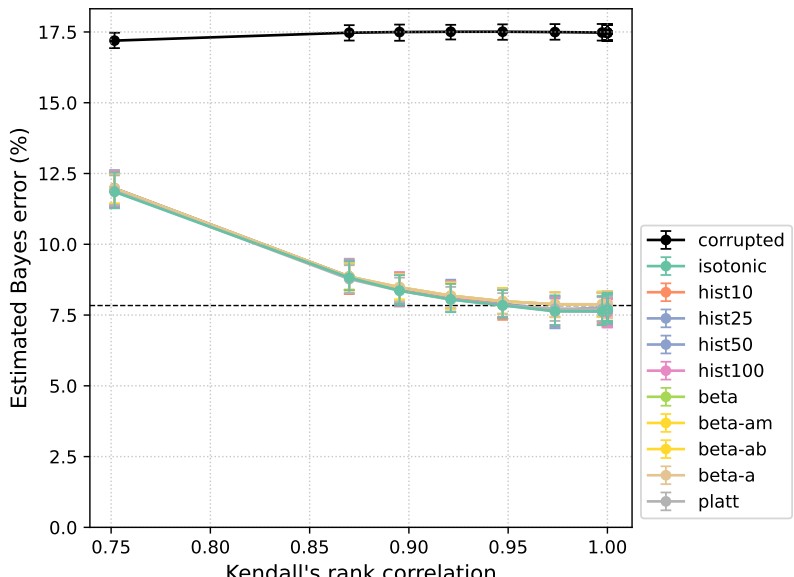

(a) Non-order-preserving corruption without additional noise.

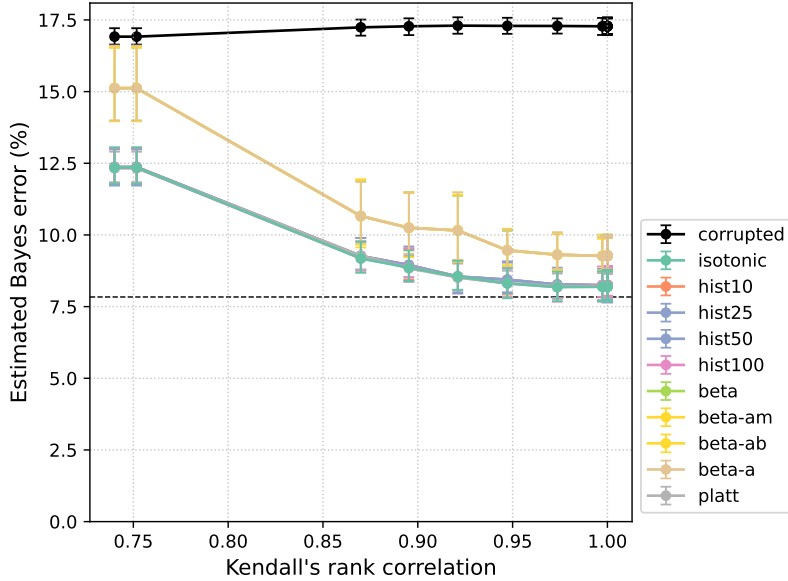

(b) Non-order-preserving corruption with additional noise, i.e., the case where corrupted soft labels are obtained by averaging $m = 50$ independent hard labels sampled from posteriors skewed by the non-order-preserving corruption.

Figure 12: Kendall tau and order breakage on synthetic logit Gaussian datasets with and without binomial noise.

