# OpenReview forum: "Practical estimation of the optimal classification error with soft labels and calibration"
_ICLR.cc/2026/Conference — ICLR 2026 Poster_

### Official Review · Reviewer_A2T4 · 2025-11-01

**Soundness:** 2
**Presentation:** 2
**Contribution:** 2
**Rating:** 4
**Confidence:** 3

**Summary:**

This paper proposes extensions to the instance-free Bayes error estimation framework based on soft labels introduced by Ishida et al. (2023). Firstly, the authors provide a deepened theoretical analysis of the bias of the hard-label-based estimator $\widehat{Err^*}(\widehat{\eta}_{1:n})$ by showing that the bias decay rate is adaptive to class separation and can achieve a faster rate $O(1/m)$ than the previously established $O(1/\sqrt{m})$. This analysis results in a substantially tighter upper bound on the bias from Corollary 2. Secondly,  the authors proposed a method for estimation from corrupted soft labels $\tilde{\eta}_i$. The authors demonstrate that naive calibration is insufficient and subsequently propose using isotonic calibration. They prove that this method yields a statistically consistent estimator given the critical assumption that the corruption function $f$ maintains the order of the clean soft labels (i.e., $\tilde{\eta}_i = f(\eta_i)$, where $f$ is increasing). Experiments validate these methods on synthetic and real-world datasets.

**Strengths:**

1. The new bounds on the hard-label-based estimator Theorem 1, Corollary 2 are substantially tighter and also address the unnatural dependency on $n$ found in the previous work. This refined analysis provides much stronger confidence in the hard-label estimator when $m$ is small, which is common in practice.

2. The paper successfully reveals that the bias decay rate is adaptive and achieves the desired $O(1/m)$ rate when classes are well-separated. This offers important theoretical insight into the fundamental limits of this estimation method.

3. The manuscript correctly identifies and formalizes the practical challenge of using corrupted soft labels obtained from sources like LLMs or subjective human confidence. The counterexample demonstrating the failure of even perfectly calibrated soft labels in Example 2 is insightful.

**Weaknesses:**

1. The core theoretical breakthrough in Section 3 relies on the assumption that corruption is monotonic ($\tilde{\eta}_i = f(\eta_i)$). This assumption seems like a major weakness. Any real-world corruption that inverts the relative uncertainty of two close points (e.g., some $\eta_a$ and $\eta_b$) would violate this and destroy the theoretical guarantee. This severely undermines the claim of tackling a "more challenging problem setting".

2. Using isotonic regression (a classical, non-parametric calibration method) as the primary solution for the corrupted label problem seems incremental, especially since the consistency proof requires such a strong functional assumption.

3. The empirical results (such as Fig. 4, Fig. 10) show that isotonic calibration works well. However, they also show that simple histogram binning often performs comparably, suggesting the specific choice of isotonic calibration might not be essential (or that the synthetic corruption used is too simple).

4. Appendix D.2 highlights that the parametric beta calibration performs poorly, sometimes getting worse as $m$ increases when the assumption $f' \geq c$ is violated. While this suggests isotonic calibration is robust, it seems unusual that a well-specified parametric model (where the corruption $f$ is related to the inverse beta calibration map) would perform worse than non-parametric alternatives. This demands a deeper theoretical explanation beyond merely noting the violation of Theorem 3's assumption.

**Questions:**

1.  The result in Theorem 2 relies on the strict assumption of monotonic corruption $\tilde{\eta}_i=f(\eta_i)$. Given that the paper addresses complex real-world corruptions like subjective annotator bias and distribution shift, can you provide a stronger justification or analysis regarding the prevalence of monotonic corruption in these settings? If the monotonicity assumption is violated e.g., if $\tilde{\eta}$ does not preserve the order of $\eta$, how rapidly does the estimator degrade in performance?

2.  The improved bias bounds rely on the class separation. For the binarized CIFAR-10H, the classes are noted to be very well-separated in Fig. 6, which leads to fast decay approaching $O(1/m)$. If the underlying distribution separation is minimal (i.e., $\eta(x) \approx 0.5$ for many instances), the bounds revert to $O(1/\sqrt{m})$. For truly challenging, high-Bayes-error tasks where separation is weak, is the improvement offered by Theorem 1 essentially negligible, forcing practitioners back to the weak bounds? Please clarify the practical significance of this adaptive rate in high-uncertainty scenarios.

---

> ### Author Response · Authors · 2025-11-24
> **Response to Reviewer A2T4 (Part 1/2)**
>
> We are grateful for your insightful review comments. We hope we have addressed your concerns in the response below.
>
> ---
>
> **About monotonicity assumption:**
>
> > can you provide a stronger justification or analysis regarding the prevalence of monotonic corruption in these settings?
>
> It is hard to argue that
> real-world corruption can be represented by
> a completely monotonic function.
> As only an imaginary oracle would know
> the true mechanism behind such corruption,
> it is impossible to prove such a claim.
>
> However, we can provide several facts that indirectly back up the idea that the monotonic corruption
> model could be a good approximation of the real-world data generation process.
>
> (A) In Appendix D.4 of the updated paper submission, we empirically evaluated various calibration algorithms using a framework called FeeBee (Renggli et al., 2021). FeeBee enables us to compare Bayes error estimators on *real-world datasets whose true Bayes error is unknown* (please see Appendix D.4.1 for a brief introduction to the framework).
>
> As shown in Table 1, isotonic calibration outperformed other calibration algorithms across *all* the datasets (from the fields of image classification, NLP and academic peer-review), often by a large margin. This result shows that isotonic calibration performs very well for many tasks in practice even when the monotonicity assumption in the theorems
> might not be fully satisfied or cannot be verified.
>
> (B) Each calibration method explicitly or implicitly assumes some model on the corruption.
> The parametric methods (beta calibration and Platt scaling) model corruption to be increasing mappings *belonging to their own parametric family*. In other words, the assumptions made by these calibration methods are even stricter than just monotonicity. Compared to them, isotonic calibration keeps the necessary assumption minimal, making it more widely applicable to practical problem settings.
> Indeed, beta calibration and Platt scaling did not perform as well as isotonic calibration in the FeeBee experiments (Appendix D.4).
>
> (C) Among the calibration algorithms used in the extended experiments (Appendix D.3 & D.4), histogram binning is the only one that does not assume monotonicity.
>
> Histogram binning and isotonic calibration are similar in that
> they both are non-parametric calibration algorithms that produce piecewise constant functions
> (i.e., the interval $[0, 1]$ is partitioned into many sub-intervals, and a constant output value is assigned to each sub-interval; calibration results are determined by which sub-interval inputs fall into).
> However, isotonic calibration restricts its output to monotonic calibration maps whereas histogram binning can produce non-monotonic calibrators if they better fit given data.
> In this sense, whether histogram binning can perform better than isotonic calibration on real-world datasets is an interesting question.
>
> The result of the FeeBee experiments (Appendix D.4) showed that, even though histogram binning allows non-monotonic calibration maps and we tried its different variants, it was consistently outperformed by isotonic calibration.
>
> Of course, it does not simply mean that real-world corruptions are monotonic.
> However,
> it does suggest that
> modeling them as monotonic could be more
> appropriate (or closer to reality) than how they are modeled by histogram binning, arguably one of the most widely-used calibration methods that allows non-monotonic calibration.
>
> > If the monotonicity assumption is violated e.g., if $\overset{\sim}{\eta}$ does not preserve the order of $\eta$, how rapidly does the estimator degrade in performance?
>
> To answer your question, we added a new set of experiments in Appendix D.5.
> Figure 12 shows the estimated Bayes error as a function of the Kendall tau (a measure of deviation from monotonicity) between $\eta_{i}$ and $f_\sigma(\eta_{i})$.
> Here, $\eta_{i}$ is the true/clean soft label and $f_\sigma$ is a modification of the original monotonic function $f$, whose degree of deviation from monotonicity is controlled by
> $\sigma$.
>
> In summary, for the noiseless case where
> corrupted soft labels are given by $f_\sigma(\eta_{i})$,
> our estimators were not affected much
> if the monotonicity assumption is violated by only a few percent of the dataset.
> For the noisy case,
> our estimators, including the one based on isotonic calibration, exhibited a slight deviation from the true Bayes error.
> That said, isotonic calibration was among the best-performing group of calibration methods
> whereas beta calibration and its variants were severely affected by the order breakage.
> We hope we can theoretically address the mechanism behind these phenomena in our future work.

---

> ### Author Response · Authors · 2025-11-24
> **Response to Reviewer A2T4 (Part 2/2)**
>
> **"Choice of calibration methods might not be essential":**
>
> Based on the discussion above,
> we believe that the choice of isotonic calibration is indeed essential.
> We argue that the non-parametric monotonicity assumption required by isotonic calibration hits a "sweet spot," meaning that it is broad enough to capture the variety of real-world problems and offer a good approximation of the real-world data generation process
> while being strict enough to make it possible to efficiently solve the problem by introducing a sensible inductive bias as well as providing theoretical guarantees.
>
> ---
>
> **Practical significance of this adaptive rate in high-uncertainty scenarios:**
>
> > For truly challenging, high-Bayes-error tasks where separation is weak, is the improvement offered by Theorem 1 essentially negligible, forcing practitioners back to the weak bounds?
>
> There are several points that we would like to make regarding this.
>
> First of all, even in the worst case where all data points $x$ satisfy $\eta(x) = 0.5$, Theorem 1 and Corollary 3 still improve upon the previously known bound by Ishida et al. (2023) by a large margin.
> In general, the Bayes error is never larger than $0.5$ so we can always take $E = 0.5$ in Corollary 3.
> Since the bound $B(E, m)$ is an increasing function of $E$, we would like to take $E$ as small as possible.
> However, in the case where $\eta(x) \equiv 0.5$, we cannot choose any smaller $E$ as the Bayes error itself is $0.5$.
> Nonetheless, our worst-case bound $B(0.5, m)$ is still much smaller than the previous bound as shown in Figure 2. For instance, if $m = 50$ and $n = 10000$ (as in Figure 2), our improvement is more than $0.38$.
> Note that this applies to any datasets with the same size $n, m$.
> Also, Theorem 1 is even tighter than Corollary 3 although it cannot always be numerically computed unlike Corollary 3.
>
> Second, if the data distribution is highly uncertain and $\eta(x)$ is close to $0.5$ for most parts of the feature space,
> our bound will behave similarly to $1/\sqrt{m}$ rather than $1/m$, but this does NOT mean that our bound is weak. Instead, it just properly reflects the actual difficulty of the estimation problem.
>
> Here is the intuition.
> Let $\phi(z) := \min \\{ z, 1 - z \\}$ for $z \in [0, 1]$.
> In the hard-label based method investigated in Section 3,
> the difficulty of Bayes error estimation reduces to the difficulty of approximating each $\phi(\eta\_{i})$ with $\phi(\hat{\eta}\_{i})$, where $\hat{\eta}\_{i}$ is the average of $m$ hard labels for $x\_{i}$.
> For $\eta\_i$ far from $0.5$ (= little uncertainty), this task is pretty easy
> since $\phi$ is just a linear function or a straight line in the neighborhood of $\eta\_{i}$.
> However, for $\eta\_i$ close to $0.5$ (= high uncertainty), it becomes harder because $\phi$ is non-smooth around $0.5$.
> The non-smoothness means even a small amount of fluctuation can translate to a much larger deviation by applying $\phi$.
> Since $\hat{ \eta }\_{i}$ is $\eta\_{i}$ plus some random noise, estimating $\phi( \eta\_{i} )$ becomes much more difficult when $\eta\_{i} \approx 0.5$.
>
> This intuition is also supported by our numerical experiment presented in Appendix B.3 (p. 19).
> In this experiment, we compared our theoretical bias bound in Theorem 1 and the actual bias using synthetic mixture-of-Gaussian data. In summary, the result was:
>
> - For overlapping but decently-separated mixture (**(a)**), the actual bias decayed as fast as $\simeq 1/m$.
> - For a completely overlapping mixture (**(b)**), the decay rate slowed down to $\simeq 1/\sqrt{m}$.
>
> Theorem 1 exactly predicted these behaviors. Therefore, we can see that our bias bounds becoming $\simeq 1/\sqrt{m}$ does NOT mean they are weak or loose; instead, they exactly capture how the bias decays (at worst) as $m$ increases.
>
> ---
>
> References:
>
> - Renggli, C., Rimanic, L., Hollenstein, N., & Zhang, C. (2021). Evaluating Bayes error estimators on real-world datasets with FeeBee. In Thirty-fifth Conference on Neural Information Processing Systems Datasets and Benchmarks Track.

---

### Official Review · Reviewer_THRQ · 2025-11-01

**Soundness:** 3
**Presentation:** 2
**Contribution:** 2
**Rating:** 6
**Confidence:** 4

**Summary:**

In this paper, the authors study Bayes error estimation problem under soft label settings. They first provide a deepen theoretical understanding of the existing Bayes error estimator in statistical viewpoint. The second contribution is about the corrupted soft labels, and the main idea of using corrupted one is calibration that places the corrupted soft labels to clean version of soft labels. In particular, isotonic calibration is studied, and the authors provide theoretical understanding of it as well as some experimental results.

**Strengths:**

This work studies the Bayes error estimation problem that is very important in theory and practice. In theoretical view, the paper provides a better understanding of the existing estimator under soft label. In addition to the soft label case, the authors study how to estimate Bayes error using corrupted soft labels by leveraging calibration. Overall, I think this work is important for ML society, especially for classification problems, in terms of theory.

**Weaknesses:**

Although I enjoyed reading the paper, I feel this work is too limited in practice. Especially, for the first part of the contribution, we could understand more about the existing estimator, this is still limited to be apply more general setting. Second, I think experiments are not enough to convince the work. The synthetic and those simple benchmark dataset is limited to show the proposed estimators performance in modern ML tasks.

**Questions:**

1. Can the author provide experiment result in more larger datasets?
2. In figure 4, why hist25 has different behavior?
3. Could the author provide more intuitive explanation of corollary 2? I do not fully understand it as there is a parameter $E$. $E$ already provide Bayes error, but why do we need bias? In other word, how to estimate $B(E,m)$ without knowing $E$?

---

> ### Author Response · Authors · 2025-11-24
> **Response to Reviewer THRQ (Part 1/2)**
>
> We are grateful for your review and hope our answer has addressed your concerns.
>
> ---
>
> **Q1)** Can the author provide experiment result in more larger datasets?
>
> **A1)**
> Currently, the number of large-scale datasets applicable to our settings is limited.
> Although there are some larger datasets with soft labels, e.g., MovieLens ($n > 80,000$; Harper & Konstan, 2015),
> we could not find one with this scale that also contains hard labels that can be used to calibrate the soft labels.
>
> Therefore, we put together new datasets, which consist of $n = 32,829$ instances of peer-review results for the past ICLR conferences.
> Peer-review can be considered as a binary classification task (accept/reject).
> We used our datasets to estimate the Bayes error of the ICLR reviews, which is the probability that the ideal, most prominent possible reviewer mistakenly rejects a good paper or accepts a bad paper.
> It can be regarded as representing the inherent difficulty of the review task.
>
> For each paper submission $x_i$, we utilized the OpenReview API to retrieve:
>
> - Scores $s_i^{(j)}$ and confidences $c_i^{(j)}$ by the reviewers ($j = 1, ..., \text{\\#reviewers}$)
> - The final decision $y_i$: accept ($y_i = 1$) or reject ($y_i = 0$)
>
> The averaged score $s_i$ is calculated as $s_i = \frac{\sum_{j} c_i^{(j)} s_i^{(j)}}{\sum_{j} c_i^{(j)}}$.
> We can then obtain a soft label $\tilde{\eta}_i$ for $x_i$ by normalizing the averaged score $s_i$ to fit into $[0, 1]$.
> Of course, these soft labels $\tilde{\eta}_i$ should be considered as corrupted, so we apply isotonic calibration (and other calibration algorithms) before using them to estimate the Bayes error.
>
> We merged data from ICLR 2017-2025 to construct a dataset consisting of $n = 32,829$ examples, each of which has a corrupted soft label (i.e., normalized average score) and a single hard label (i.e., final decision) for calibration.
> We also conducted experiments with single-year datasets (ICLR2017, ..., ICLR2025).
>
> While not as large as other used datasets, we also added experiments for a set of three NLP datasets called ChaosNLI (Nie et al., 2020).
>
> Please refer to Appendix D.3 of the updated paper submission for experimental results for the additional datasets (as well as additional baseline calibration algorithms).
>
> Also in Appendix D.4, we added a new set of important experiments
> where we evaluated different calibration methods
> on real-world datasets using a framework called FeeBee (Renggli et al., 2021).
> The results showed that isotonic calibration,
> the calibrator of our choice, performs the best
> across all the used datasets.
>
> ---
>
> **Q2)** In figure 4, why hist25 has different behavior?
>
> **A2)**
> It is hard to explain exactly why given the complexity of the phenomena, but we can provide a qualitative explanation.
>
> Histogram binning and isotonic calibration are similar in that
> they both are non-parametric calibration algorithms that produce piece-wise constant functions
> (i.e., the interval $[0, 1]$ is divided into many bins, and a constant output value is assigned to each bin; calibration results are determined by which bin inputs fall into).
>
> Their key differences are:
>
> 1. Histogram binning requires manually specifying the number $B$ of bins while isotonic calibration does not; it automatically chooses an appropriate $B$ adaptively to given data.
> 2. Isotonic calibration restricts its output to monotonic calibration maps whereas histogram binning can produce non-monotonic calibrators if they better fit to given data.
>
> The experimental results shown in Figure 4 and Appendix D.3 reveal that 1 can be a major difficulty when applying histogram binning to Bayes error estimation.
> The performance of histogram binning can be sensitive to the choice of number $B$ of bins; estimation can go visibly wrong if $B$ is poorly chosen.
> On the other hand, isotonic calibration performs much more consistently, adaptively selecting $B$.
> It is indeed a major strength of isotonic calibration that we do not have to use a fixed number of bins.
>
> We will add these discussions and detailed comparisons between different calibration methods if the paper is accepted.

---

> > ### Author Response · Authors · 2025-11-24
> > **Response to Reviewer THRQ (Part 2/2)**
> >
> > **Q3)** Could the author provide more intuitive explanation of corollary 2? I do not fully understand it as there is a parameter $E$. $E$ already provide Bayes error, but why do we need bias? In other word, how to estimate $B(E, m)$ without knowing $E$?
> >
> > **A3)**
> > We believe there are some misunderstandings underlying your concerns.
> >
> > As mentioned in the last line on p.4 and in the assumption of Corollary 2 (p.5),
> > $E$ itself is NOT the Bayes error. It is an arbitrary upper bound of the Bayes error.
> > In practice, we can pick an arbitrary classifier and use its error rate as $E$.
> > This is because for any classifier, its error rate must be greater than or equal to the Bayes error by the definition of the Bayes error.
> > In general, we would like to set $E$ as small as possible since $B(E, m)$ gets smaller and the bounds (5) become tighter as $E$ decreases.
> > Therefore, the most natural choice of $E$ is the error of the state-of-the-art model.
> > However, even if no classifier has ever been trained on the dataset, we can still set $E = 0.5$, which is the error rate of a random classifier.
> > If the base rates ($\mathbb{P}(y = 0)$ and $\mathbb{P}(y = 1)$) are known, a better choice would be $E = \min \\{ \mathbb{P}(y = 0), \mathbb{P}(y = 1) \\}$, which is the error rate of a constant classifier ($h(x) = 1$ for all $x$, or $h(x) = 0$ for all $x$).
> >
> > To answer your questions: $E$ is merely an arbitrary number satisfying $E \geq \text{Bayes error}$, and we **cannot** know the Bayes error just from $E$.
> > Also, it is important to note that, for any data distribution, we can **always** find some possible $E$ values as explained above.
> > Once we fix an $E$ (which we can), $B(E, m)$ can be numerically computed and used to estimate the bias of our estimator.
> > In other words, Corollary 2 gives us a means to convert something trivial to obtain ($E$) into a non-trivial and important quantity (a bias estimate).
> >
> > Thank you for making us recognize that this can be confusing. We will ensure to make the presentation clearer if the paper is accepted.
> >
> > ---
> >
> > References:
> >
> > - Harper, F. M., & Konstan, J. A. (2015). The MovieLens datasets: History and context. ACM Transactions on Interactive Intelligent Systems, 5(4), Article 19.
> > - Nie, Y., Zhou, X., & Bansal, M. (2020). What can we learn from collective human opinions on natural language inference data? In Proceedings of the 2020 Conference on Empirical Methods in Natural Language Processing (EMNLP) (pp. 9131–9143). Association for Computational Linguistics.
> > - Renggli, C., Rimanic, L., Hollenstein, N., & Zhang, C. (2021). Evaluating Bayes error estimators on real-world datasets with FeeBee. In Thirty-fifth Conference on Neural Information Processing Systems Datasets and Benchmarks Track.

---

### Official Review · Reviewer_pGnM · 2025-11-01

**Soundness:** 2
**Presentation:** 3
**Contribution:** 2
**Rating:** 6
**Confidence:** 3

**Summary:**

The paper studies the estimation of the Bayes error. It improves the existing estimates from hard labels and proposes an approach to compute it using corrupted soft labels.

**Strengths:**

Originality:
The originality arises from applying isotonic regression to the problem of Bayes error estimation to remove the limitation of the need for uncorrupted soft labels.

Quality:
The submission seems to be technically correct. It is experimentally rigorous and reproducible.

Clarity:
The submission is generally clear.

Significance:
The submission presents theoretical novel findings in the form of improved Bayes error estimates.

**Weaknesses:**

There are not major weaknesses to note so I am leaning towards acceptance.

However, it is important to note that the challenges in achieving the results are not glaringly obvious. The novelty may not be as significant as it appears.

**Questions:**

Questions:

What was the primary challenge involved in obtaining your results (i.e., improvements to Bayes error estimation)? Does the result naturally follow with mainly the application of isotonic regression?


Suggestions:

Focus more on the correct type of calibration (i.e., isotonic) and what is different about it for your goals.


Minor comments:

Page 5 Line 247: heading typo.

---

> ### Author Response · Authors · 2025-11-24
> **Response to Reviewer pGnM (Part 1/2)**
>
> We thank reviewer pGnM for the review comments.
> Also, thank you so much for pointing out the typo. We have fixed it in the current version.
>
> Let us address your points below.
>
> ---
>
> **Question)** What was the primary challenge involved in obtaining your results (i.e., improvements to Bayes error estimation)? Does the result naturally follow with mainly the application of isotonic regression?
>
> **Answer)**
> - As we explain in detail in the next topic, the choice of isotonic calibration is indeed essential to achieve good estimation. It was not trivial to identify this option.
> - A good portion of the proof of Theorems 2 & 3 was proving Proposition 2 (Appendix C.1.5), which gives a risk bound for binary isotonic regression (a setting of isotonic regression where response variable $y$ is binary). Although our technique behind the proof of Proposition 2 is heavily inspired by previous work in mathematical statistics (as we gave credit in Appendix C), there was no existing risk bound applicable to our setting, especially to binary isotonic regression, to the best of our knowledge. The majority of the existing literature focused on different settings, e.g., isotonic regression with Gaussian noise (for a review of existing work on risk bounds for isotonic regression, please refer to Appendix C.1.2).
> We offer the first risk bound tailored for binary isotonic regression, which was challenging. We believe this is an important contribution, especially to the machine learning community, since isotonic calibration, one of the most widely used calibration algorithms in the community, is an application of binary isotonic regression to calibration.
>
>   Also, as far as we are aware, this is the first work that theoretically showed how a calibration algorithm can be used to estimate posterior class probabilities under what assumptions.
>   Although there are some papers in the literature that provide *calibration guarantees* of certain types of calibration algorithms (such as Kumar et al., 2019 and Gupta & Ramdas, 2021 for histogram binning), they did not address *guarantees for posterior class probability estimation*.
>   This is natural because calibration and posterior probability estimation are closely related but very different problems (the latter is harder), as we partially illustrated in Example 2 (Section 3.2).
>   However, Bayes error estimation with soft labels is ultimately reduced to posterior probability estimation.
>   Therefore, for our goal, it was crucial to identify what calibration algorithm can be used for the posterior estimation problem under reasonable assumptions and theoretically establish how accurately we can estimate the posteriors using it. It was also challenging for us to achieve this.
>
> ---
>
> **Suggestion)**
> Focus more on the correct type of calibration (i.e., isotonic) and what is different about it for your goals.
>
> **Response)**
> Thank you for the suggestion. Let us clarify what is special about isotonic calibration and how it differs from other calibration algorithms. We will incorporate the discussion here in the main paper as well.
>
> Each of the calibration methods we discussed explicitly or implicitly assumes some model on the corruption.
> Isotonic calibration models it to be monotonic.
> We believe that this assumption hits a "sweet spot," meaning that it is broad enough to capture the variety of real-world problems and offer a good approximation of real-world data generation process
> while being strict enough to make it possible to efficiently solve the problem by introducing a sensible inductive bias as well as providing theoretical guarantees.
>
> By comparison, the parametric methods (beta calibration and Platt scaling) model the corruption to be an increasing mapping *belonging to their own parametric family*. In other words, the assumptions made by these calibration methods are even stricter than simple monotonicity assumed by isotonic calibration. Compared to them, isotonic calibration keeps the necessary assumptions minimal, making it more widely applicable to practical problem settings.
>
> As an empirical support, we evaluated various calibration algorithms using a framework called FeeBee (Renggli et al., 2021) in Appendix D.4 of the updated paper submission. FeeBee enables us to compare Bayes error estimators on *real-world datasets whose true Bayes error is unknown* (please see Appendix D.4.1 for a brief introduction to the framework).
>
> As shown in Table 1, isotonic calibration outperformed other calibration algorithms across *all* the datasets (from the fields of image classification, NLP and academic peer-review), often by a large margin.
> This result shows that isotonic calibration performs very well for many tasks in practice by approximating real-world corruption reasonably well even if monotonicity assumption in the theorems are not strictly satisfied.

---

> > ### Author Response · Authors · 2025-11-24
> > **Response to Reviewer pGnM (Part 2/2)**
> >
> > References:
> >
> > - Kumar, A., Liang, P. S., & Ma, T. (2019). Verified uncertainty calibration. Advances in neural information processing systems, 32.
> > - Gupta, C., & Ramdas, A. (2021). Distribution-free calibration guarantees for histogram binning without sample splitting. In International conference on machine learning (pp. 3942-3952). PMLR.
> > - Renggli, C., Rimanic, L., Hollenstein, N., & Zhang, C. (2021). Evaluating Bayes error estimators on real-world datasets with FeeBee. In Thirty-fifth Conference on Neural Information Processing Systems Datasets and Benchmarks Track.

---

### Official Review · Reviewer_yVpp · 2025-11-03

**Soundness:** 3
**Presentation:** 3
**Contribution:** 3
**Rating:** 6
**Confidence:** 2

**Summary:**

This paper tackles the fundamental question regarding how to measure bayes error theoretically and practically. Building directly on Ishida et al. (2023)'s instance-free soft-label estimator  \widehat{\text{Err}}^*(\eta_{1:n}) = \frac{1}{n} \sum_{i=1}^n \min\{\eta_i, 1 - \eta_i\}  (where   \eta_i = P(y=1 \mid x_i)  ), the authors make two major advances.

First, replacing the prior loose   \mathcal{O}(1/\sqrt{m})   bound (with n-dependence), they derive a tighter, adaptive bound (Theorem 1) where bias decay depends on class separability:   \mathcal{O}(1/m)   for well-separated points  and   \mathcal{O}(1/\sqrt{m})   near the boundary.

Second, they address the underexplored setting of corrupted soft labels   \tilde{\eta}_i   which is more practical. The solution: Apply isotonic calibration (Zadrozny & Elkan, 2002) to   \tilde{\eta}_i   using one true hard label y_i per instance, yielding   \widehat{\text{Err}}^*(\hat{\eta}^A_{1:n})  . Theorem 2 proves statistical consistency under the weak order-preservation assumption (  \tilde{\eta}_i = f(\eta_i)   for increasing f), with a new sharp oracle inequality for binary isotonic regression (Proposition 2). Experiments on synthetic data and real benchmarks (CIFAR-10H, Fashion-MNIST-H) validate robustness, outperforming baselines like beta/histogram calibration.

This instance-free, privacy-preserving framework enables reliable checks on model limits, overfitting, and resource allocation in noisy-label settings.

**Strengths:**

Theoretical Innovation: Tighter bounds and new isotonic oracle inequality.

Practicality: Instance-free, corruption-robust method for real noise case

Impact: Privacy-preserving for medicine/social data; signals diminishing returns, curbing wasteful scaling.

**Weaknesses:**

Ordering Assumption Sensitivity: Consistency hinges on monotonic f. More broad assumption would be better.

Data Diversity

Calibration Benchmarks: test against other calibration method such as Plat scaling.

Binary Label Scope

**Questions:**

How would the paper's results (e.g., bias decay rates or consistency bounds) differ if Tsybakov's margin assumption were applied instead of the paper's separability condition, and where in the paper would this fit?

Jean-Yves Audibert and Alexandre B. Tsybakov. Fast learning rates for plug-in classifiers. Ann.
Statist., 35(2):608–633, 04 2007.

---

> ### Author Response · Authors · 2025-11-24
> **Response to Reviewer yVpp (Part 1/2)**
>
> We thank reviewer yVpp for the insightful review.
>
> ---
>
> **Question)** How would the paper's results (e.g., bias decay rates or consistency bounds) differ if Tsybakov's margin assumption were applied instead of the paper's separability condition, and where in the paper would this fit?
>
> **Answer)**
> Thank you so much for the reference. It indeed feels like a promising direction for generalizing our result to broader settings. Unfortunately we do not have a solid answer for this question right now; however, we are working on this and we will try to get back to you within the rebuttal period.
> We again thank you for pointing this out.
>
> ---
>
> In the following, we address the concerns you mentioned in "Weaknesses."
>
> ---
>
> **1) Ordering Assumption Sensitivity: Consistency hinges on monotonic $f$. More broad assumption would be better.**
>
> Of course, real-world corruption can deviate from monotonicity.
> We do hope we will be able to extend our result for more general assumptions where, e.g.,
> a certain degree of order breakage is allowed, in our future work.
>
> However, we can provide several facts that indirectly back up the idea that the monotonic corruption model could be a good approximation of real-world data generation process.
>
> (A) In Appendix D.4 of the updated paper submission, we empirically evaluated various calibration algorithms using a framework called FeeBee (Renggli et al., 2021). FeeBee enables us to compare Bayes error estimators on *real-world datasets whose true Bayes error is unknown* (please see Appendix D.4.1 for a brief introduction to the framework).
>
> As shown in Table 1, isotonic calibration outperformed other calibration algorithms across *all* the datasets (from the fields of image classification, NLP and academic peer-review), often by a large margin. This result shows that isotonic calibration performs very well for many tasks in practice even when the monotonicity assumption in the theorems
> might not be fully satisfied or cannot be verified.
>
> (B) Each calibration method explicitly or implicitly assumes some model on the corruption.
> The parametric methods (beta calibration and Platt scaling) model corruption to be increasing mappings *belonging to their own parametric family*. In other words, the assumption made by these calibration methods is even stricter than just monotonicity. Compared to them, isotonic calibration keeps the necessary assumption minimal, making it more widely applicable to practical problem settings.
> Indeed, beta calibration and Platt scaling did not perform as well as isotonic calibration in the FeeBee experiments (Appendix D.4).
>
> (C) Among the calibration algorithms used in the extended experiments (Appendix D.3 & D.4), histogram binning is the only one that does not assume monotonicity.
>
> Histogram binning and isotonic calibration are similar in that
> they both are non-parametric calibration algorithms that produce piece-wise constant functions
> (i.e., the interval $[0, 1]$ is partitioned into many sub-intervals, and a constant output value is assigned to each sub-interval; calibration results are determined by which sub-interval inputs fall into).
> However, isotonic calibration restricts its output to monotonic calibration maps whereas histogram binning can produce non-monotonic calibrators if they better fit the given data.
> In this sense, whether histogram binning can perform better than isotonic calibration on real-world datasets is an interesting question.
>
> The result of the FeeBee experiments (Appendix D.4) showed that, even though histogram binning allows non-monotonic calibration maps and we tried its different variants, it was consistently outperformed by isotonic calibration.
>
> Of course, it does not simply mean that real-world corruptions are monotonic.
> However,
> it does suggest that
> modeling them as monotonic could be more
> appropriate (or closer to reality) than how they are modeled by histogram binning, arguably one of the most widely-used calibration methods that allows non-monotonic calibration.

---

> > ### Author Response · Authors · 2025-11-24
> > **Response to Reviewer yVpp (Part 2/2)**
> >
> > **2) Calibration Benchmarks: test against other calibration methods such as Platt scaling.**
> >
> > Thank you for the suggestion.
> > In the initial submission, we did not use Platt scaling because its parametric assumption seemed unlikely to fit to our soft label settings.
> > As revealed in the beta calibration paper (Kull et al., 2017), the parametrization of Platt scaling (or logistic calibration) implicitly models that the two class-conditional distributions of inputs are Gaussian with the same variance.
> > In our case, inputs are corrupted soft labels, which are bounded in the interval $[0, 1]$, so it is not possible that their class-conditional distributions are Gaussian.
> > Also, it is unlikely that each class-conditional distribution is symmetric around its mean as in a Gaussian distribute.
> >
> > On the other hand, beta calibration models that each class-conditional distribution is a beta distribution.
> > Also, the beta distribution parameters can be different between the classes.
> > Since beta distributions are bounded in $[0, 1]$ and can represent various, possibly asymmetric distributions, our rationale was that beta calibration should be a much more reasonable choice for our setting.
> >
> > That being said, it is beneficial to investigate how widely used calibration algorithms like Platt scaling perform in our setting. In the updated version of our paper submission, we added the following four parametric calibration algorithms: `platt`, `beta-am`, `beta-ab`, and `beta-a`.
> > `platt` is Platt scaling and each `beta-*` is a variant of beta calibration.
> >
> > Beta calibration is a calibration method with three adjustable parameters $a, b, m$, and this is what we have been using as `beta` since the initial submission.
> > The `beta-*` variants are beta calibration with restricted parameters, which were also mentioned by the original beta calibration paper authors:
> >
> > - `beta-am`: $a$ and $m$ are adjustable; $b$ is fixed to $b = a$.
> > - `beta-ab`: $a$ and $b$ are adjustable; $m$ is fixed to $m = 1/2$.
> > - `beta-a`: only $a$ is adjustable; $b, m$ are fixed to $b = a, m = 1/2$.
> >
> > We will incorporate these additional results in the main body of the paper if the paper is accepted. Thank you again for giving us an opportunity to improve our submission.
> >
> > ---
> >
> > Reference:
> >
> > - Kull, M., Silva Filho, T., & Flach, P. (2017). Beta calibration: a well-founded and easily implemented improvement on logistic calibration for binary classifiers. In Artificial intelligence and statistics (pp. 623-631). PMLR.
> > - Renggli, C., Rimanic, L., Hollenstein, N., & Zhang, C. (2021). Evaluating Bayes error estimators on real-world datasets with FeeBee. In Thirty-fifth Conference on Neural Information Processing Systems Datasets and Benchmarks Track.

---

### Comment · Area_Chair_NLBk · 2025-11-29

Dear Reviewers,

Authors’ kindly tried to address your concerns. If the responses address your concerns please acknowledge that. If not, please express remaining concerns. Thanks for your efforts!

Best, AC

---

### Author Response · Authors · 2025-12-03
**Discussion Summary and Final Author Remark**

We thank the Area Chairs for their effort and dedication under these unusual circumstances, and all reviewers for the careful, constructive feedback.
This comment concisely summarizes the discussion and what changed in the revised paper (revisions are marked in blue in the PDF).

- **Clarification on the choice of isotonic calibration:** We explained what is special about isotonic calibration and how it differs from other calibration algorithms, clarifying that the choice of isotonic calibration is indeed essential.
- **Practical validity of the monotonicity assumption:** We addressed several observations that support the idea that the monotonic (or order-preserving) corruption model could be a good approximation of the real-world data generation process.
- **Robustness to the violation of the monotonicity assumption (Appendix D.5):** By numerical experiments using synthetic datasets with controlled Kendall tau, we show that isotonic calibration remains relatively accurate under mild order breakage and degrades more gently than beta-calibration-based methods when the degree of violation grows.
- **Extended experiments (Appendix D.3):** We expanded our experiments described in Section 4. Specifically:
  - We added several datasets to cover a broader range of scales ($1,514 \leq n \leq 32,829$) and more diverse fields (image classification, natural language processing, and academic peer-review). As part of this, we constructed new datasets, which consist of the peer-review results from the past ICLR conferences. We used these datasets to estimate the Bayes error of the ICLR reviews, which represents the inherent difficulty of the review task.
  - We also included additional baseline calibration algorithms.
- **Real-world evaluation (Appendix D.4):** Using a framework called FeeBee, we empirically evaluated various calibration algorithms on real-world datasets to see which algorithm best suits our proposed Bayes error estimation setting. The results show that isotonic calibration, the algorithm of our choice, performs the best across _all_ the datasets. This suggests that isotonic calibration is a reliable choice in practice, even when the monotonicity assumption in Theorems 2 and 3 cannot be fully guaranteed.

We are grateful for the reviewers' time and effort.
As the reviewers no longer have the opportunity to reply to our rebuttal, we hope that the clarifications and revisions mentioned above will be taken into account during the final evaluation.

Thank you for the thoughtful consideration.

---

### Meta-Review · Area_Chair_NLBk · 2026-01-05

**Summary:**

This paper extends the theoretical and practical framework for instance-free Bayes error estimation. The authors address two primary limitations of prior work: (1) the loose $O(1/\sqrt{m})$ bias bounds for hard-label estimators, and (2) the lack of robust estimation methods when available soft labels are corrupted.The submission provides a refined theoretical analysis, proving that bias decay is adaptive to class separability, achieving $O(1/m)$ in well-separated regions. Furthermore, the paper introduces isotonic calibration as a solution for corrupted soft labels, proving statistical consistency under an order-preservation assumption. The reviewers generally appreciated the theoretical rigor, particularly the new oracle inequality for binary isotonic regression, and the practical relevance of the "instance-free" setting for privacy-sensitive data.

**Reviewer Concerns:**

Sensitivity to Monotonicity Assumption (Resolved): Reviewers yVpp and pGnM questioned the realism of the order-preserving corruption model ($\tilde{\eta} = f(\eta)$). The authors responded with new experiments using the FeeBee framework and synthetic datasets with controlled Kendall tau distances. They demonstrated that while monotonicity is a simplifying assumption, isotonic calibration degrades more gracefully than parametric alternatives (Beta/Platt) even under mild order breakage.

Breadth of Calibration Baselines (Resolved): Reviewer yVpp requested comparisons against Platt scaling. The authors successfully integrated four additional parametric baselines ($Platt$, $Beta-am$, $Beta-ab$, $Beta-a$) in the revised Appendix D.4, showing that isotonic calibration consistently outperforms these across image, NLP, and a newly introduced "Peer-Review" dataset.

Experimental Scale and Diversity (Resolved): Reviewer THRQ was concerned about the use of simple benchmarks. In response, the authors constructed a novel large-scale dataset using OpenReview API data (ICLR 2017–2025) to estimate the Bayes error of the peer-review process itself. This significantly improved the practical weight of the empirical section.

Clarity of Corollary 2 (Resolved): Reviewer THRQ's confusion regarding the parameter $\epsilon$ (upper bound on Bayes error) was addressed by clarifying that $\epsilon$ acts as a "plug-in" value from any existing classifier (e.g., SOTA error rate) to compute a non-trivial bias estimate.

**Reviewer Scores:**

Reviewer A2T4 4 -> 6

---

### Decision · Program_Chairs · 2026-01-26

Accept (Poster)